# White-Basilisk: A Hybrid Model for Code Vulnerability Detection

## Abstract

The proliferation of software vulnerabilities presents a significant challenge to cybersecurity, necessitating more effective detection methodologies. We introduce White-Basilisk, a novel approach to vulnerability detection that demonstrates superior performance while challenging prevailing assumptions in AI model scaling. Utilizing an innovative architecture that integrates Mamba layers, linear self-attention, and a Mixture of Experts framework, White-Basilisk achieves state-of-the-art results in vulnerability detection tasks with a parameter count of only 200M. The model's capacity to process sequences of unprecedented length enables comprehensive analysis of extensive codebases in a single pass, surpassing the context limitations of current Large Language Models (LLMs). White-Basilisk exhibits robust performance on imbalanced, real-world datasets, while maintaining computational efficiency that facilitates deployment across diverse organizational scales. This research not only establishes new benchmarks in code security but also provides empirical evidence that compact, efficiently designed models can outperform larger counterparts in specialized tasks, potentially redefining optimization strategies in AI development for domain-specific applications.

## 1 Introduction

The field of Artificial Intelligence (AI) has experienced significant advancements in recent years, particularly in the domain of Natural Language Processing (NLP). Large Language Models (LLMs) such as GPT, Llama, and Gemini have demonstrated remarkable capabilities across diverse tasks. These models, often comprising hundreds of billions of parameters, reflect the "bigger is better" philosophy in AI development. However, even with the notable success of these massive models, they present substantial challenges. For instance, the computational requirements for training and inference of such models are considerable, resulting in high energy consumption and limited accessibility (Patterson et al., 2021; Faiz et al., 2023). As these models continue to expand in size and complexity, it is important to reconsider the sustainability and necessity of this approach for all AI applications.

One domain where the limitations of the "bigger is better" paradigm become particularly evident is in specialized tasks like code vulnerability detection. This critical aspect of cybersecurity requires models that can comprehend complex code structures, identify subtle patterns, and maintain high accuracy – all while ideally being deployable in resource-constrained environments. Code vulnerability detection represents a unique challenge at the intersection of software engineering and cybersecurity. Vulnerabilities often emerge from complex interactions between various components in the codebase, making detection through standard techniques difficult. The rapid evolution of both software development practices and attack methodologies further exacerbates this challenge. As developers embrace novel paradigms such as microservices and AI-driven code generation, the potential attack surface expands in ways that can be challenging to predict.

Traditional approaches to automated vulnerability detection, including static application security testing (SAST) tools, have demonstrated both strengths and limitations in addressing security challenges. Studies have shown that SAST tools generally achieve lower vulnerability detection rates compared to newer approaches like large language models (LLMs) (Zhou et al., 2024). This lower detection rate is concerning because it means that SAST tools may miss critical security vulnera-

bilities, potentially leaving software systems exposed to exploitation and attacks. The inability to identify a significant portion of vulnerabilities undermines the effectiveness of these tools in ensuring robust software security, highlighting the need for more advanced or complementary detection methods.

Recent machine learning (ML)-based approaches have demonstrated promising results in vulnerability detection. However, many exhibit significant limitations in processing extended code sequences and comprehending complex, context-dependent vulnerabilities. These models often struggle with long-range dependencies and fail to capture the nuanced interactions between disparate code components. Moreover, their performance can be inconsistent across diverse vulnerability types and programming paradigms, limiting their efficacy in comprehensive security analyses. In response to these challenges, we present White-Basilisk: a compact but powerful model that challenges the prevailing paradigm in AI design. With approximately 200M parameters – a fraction of the size of many current state-of-the-art models – White-Basilisk demonstrates that thoughtful architecture and targeted training can yield impressive results in code vulnerability detection.

White-Basilisk's design philosophy centers on maximizing efficiency without compromising performance. This approach led to several innovations that form the core of our contributions, which can be summarized as follows:

1. **Efficient Architecture:** We propose a novel integration of Mamba layers, linear-complexity Infini-attention, and Mixture of Experts. This combination enables effective processing of long code sequences while maintaining a relatively small parameter count.

2. **Extended Context Length:** White-Basilisk can analyze sequences up to 128,000 tokens during inference. This capability facilitates whole-codebase analysis on a single GPU, potentially uncovering vulnerabilities that span multiple functions or files.

3. **Resource-Efficient Training:** Our model achieves competitive performance using a dataset of just 2 million code samples. This efficiency challenges conventional assumptions about data requirements for specialized AI tasks.

4. **Advanced Training Techniques:** We incorporated Fill-in-the-Middle (FIM) pretraining and Scale-Invariant Fine-Tuning (SIFT) to enhance model robustness and generalization. These techniques contribute to White-Basilisk's ability to perform well across diverse code vulnerability detection tasks.

5. **Comprehensive Benchmarking:** We conduct rigorous experiments to evaluate White-Basilisk's efficacy across multiple benchmark datasets, including PRIMEVUL Ding et al. (2024), BigVul Fan et al. (2020), Draper Russell et al. (2018), REVEAL Chakraborty et al. (2021), and VulDeepecker Li et al. (2018). Our compact model demonstrates competitive performance against larger counterparts, emerging as the front-runner in several key metrics.

By addressing the unique challenges of code vulnerability detection with a resource-efficient approach, White-Basilisk not only explores new possibilities in this critical domain but also raises pertinent questions about the necessity of ever-larger models in AI. In the subsequent sections, we will elucidate the details of White-Basilisk's architecture, training procedures, and performance across multiple benchmarks.

## 2 RELATED WORK

The field of automated code vulnerability detection, a cornerstone of our research, has undergone rapid evolution since its inception in the early 2000s. Driven by the growing complexity and security risks of software systems, early pioneers explored static analysis techniques Chess & McGraw (2004) and pattern matching methods Livshits & Lam (2005). While these approaches laid a foundational framework, they often encountered significant drawbacks, including high false positive rates, difficulty detecting complex vulnerabilities, and susceptibility to obfuscation techniques. Due to these limitations, they were generally unsuitable for active production environments.

As the field matured, researchers recognized the transformative potential of AI, gradually shifting from conventional practices to a new paradigm. This transition was marked by the development of

VulDeePecker Li et al. (2018), one of the first deep learning (DL)-based systems for vulnerability detection. VulDeePecker utilized code gadgets and Bidirectional Long Short-Term Memory (BiLSTM) networks to identify vulnerabilities in C/C++ code. This work demonstrated the ability of DL techniques to capture complex patterns associated with code vulnerabilities, paving the way for the development of further ML-driven solutions. However, its reliance on manually crafted features limited its generalizability. Building on this work, Russell et al. (2018) developed the Draper dataset, which provides a substantial real-world dataset specifically designed for training neural networks in the task of vulnerability detection. Their work showed the advantages of leveraging vast training data to enhance model performance, improving performance but still struggling with limited context windows that restricted the capture of long-range dependencies in code.

Following that, pre-trained LLMs emerged as a significant breakthrough in code analysis. Hanif & Maffeis (2022) proposed VulBERTa, an adaptation of the RoBERTa model for detecting code vulnerabilities, demonstrating the potential of transfer learning from natural language processing to code analysis. By fine-tuning LLMs pre-trained on extensive corpora of code, this approach quickly gained popularity due to its ability to capture latent patterns in both code structure and semantics. However, similar to many transformer-based models, it suffers from quadratic computational complexity with sequence length, constraining its applicability to large-scale projects.

Another line of research focused on models specifically tailored for code understanding. Feng et al. (2020) introduced CodeBERT, a bimodal pre-trained model for programming language and natural language. This work allowed for a deeper understanding of both code and its associated documentation, proving useful across a range of software engineering tasks, including vulnerability detection. However, while powerful, these models typically demand significant computational resources and encounter difficulties when processing extremely long sequences, which restricts their practical use in analyzing large codebases. For instance, CodeBERT was pre-trained for approximately 1000 hours on 16 interconnected NVIDIA Tesla V100 GPUs, representing a substantial energy and resource investment.

Other recent work concentrated on improving the granularity and efficiency of vulnerability detection. Such an example is LineVul (Fu & Tantithamthavorn (2022)), a transformer-based approach for line-level vulnerability prediction, enabling precise localization of vulnerabilities within codebases. While valuable for precise localization, this approach may overlook vulnerabilities that span multiple lines or functions.

Comprehensive studies by Chakraborty et al. (2021) and Ding et al. (2024) highlighted persistent challenges in the field, including data quality issues, unrealistic evaluation methods, and difficulties in handling long-range dependencies. Ding et al. (2024) showed that existing benchmarks significantly overestimate model performance, with state-of-the-art models achieving high scores on flawed datasets but failing on more realistic ones. Our work directly addresses these concerns through improved data collection, realistic evaluation metrics, and model architecture designed for long-range understanding.

Currently, existing vulnerability detection methods primarily focus on their main objectives without considering the increasing size of model parameters. This uncontrolled growth in model size has led to the development of energy-hungry models requiring dozens of cutting-edge GPUs. Most recently, research developments have been made in efficient language model design. For example, building on the work of Mamba Gu & Dao (2023), Lieber et al. (2024) introduced Jamba, which employs a Mixture of Experts (MoE) strategy, combining Mamba layers and attention mechanisms in an interleaving pattern Fedus et al. (2022). In addition, Wang et al. (2020) proposed the Linformer, which can reduce self-attention complexity to linear time using low-rank approximations, while the Transformer-XL architecture Dai et al. (2019) has been shown to effectively model long-term dependencies by introducing recurrence into self-attention.

In this paper, we introduce an efficient method for developing AI models. Our model, White-Basilisk (Figure 1), demonstrates how the integration of cutting-edge training techniques enables the deployment of comparatively small models with just 200M parameters — significantly fewer than the billions seen in some other cases. Furthermore, unlike current state-of-the-art models, our implementation was trained using only a single NVIDIA A100 40GB GPU, highlighting its resource efficiency. The increased model performance combined with reduced energy requirements for model

creation should draw the attention of the AI community. This approach could help reduce global energy consumption.

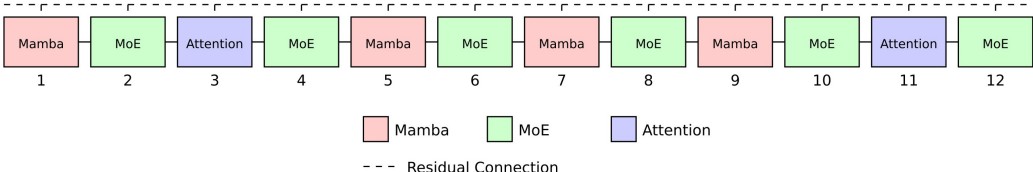

Figure 1: White-Basilisk Architecture

## 3 MODEL ARCHITECTURE

The architecture of White-Basilisk was designed to address three key challenges in code vulnerability detection. Firstly, it tackles the problem of long-range dependencies, as code vulnerabilities often span multiple functions or even files, necessitating a model capable of understanding extensive contexts. Secondly, the architecture strikes a balance between local and global information processing. This dual focus enables both fine-grained understanding of code syntax and broad comprehension of overall program structure, both crucial for effective vulnerability detection. Lastly, the design prioritizes computational efficiency, aiming to create a powerful model that can be deployed in real-world settings without requiring massive computational resources. By addressing these interconnected objectives, our model provides a robust solution for identifying and analyzing potential security flaws in code. To this end, we developed a novel hybrid architecture that combines three main components:

1. **Mamba layers:** These form the backbone of our model, efficiently capturing local dependencies and providing adaptive computation based on input content.

2. **Linear-complexity Infini-attention:** Our adaptation of this mechanism allows for efficient processing of extremely long sequences, enabling whole-codebase analysis.

3. **Mixture of Experts (MoE):** This adds dynamic adaptability throughout the network, allowing the model to specialize its processing based on input characteristics.

The synergy between these components allows White-Basilisk to process sequences up to 128,000 tokens during inference, a capability that sets it apart in the field of code analysis. This extensive context window enables holistic analysis of entire codebases, potentially uncovering vulnerabilities that span multiple functions or files. Furthermore, the layer combination pipeline, inspired by the Jamba model Lieber et al. (2024), allows for a more sophisticated interleaving pattern compared to simple alternation. Specifically, the layer combination is defined by two main configuration parameters, determined through experimentation: the attention layer offset (2) and the attention layer period (8). In addition, the conjunction of layers in our architecture is defined by the following formula:

$$\text{Layer}_i = \begin{cases} \text{Attention}(x), & \text{if } (i-2) \bmod 8 = 0 \text{ and } i \geq 2 \\ \text{MoE}(x), & \text{if } i \bmod 2 = 1 \\ \text{Mamba}(x), & \text{otherwise} \end{cases} \quad (1)$$

The forward pass through the model can be expressed as $h_i$, where $i \in \{0, 1, ..., L-1\}$ is the layer index and $L$ is the total number of layers in the architecture. Here, $h_i$ is the hidden state after the $i$-th layer, and $h_0$ is the initial input embedding. The final output of the model $y$ is obtained by applying layer normalization to the last hidden state $h_L$:

$$h_i = \text{Layer}_i(h_{i-1}) + h_{i-1} \quad and \quad y = \text{LayerNorm}(h_L) \quad (2)$$

The residual connections ($h_i = \text{Layer}_i(h_{i-1}) + h_{i-1}$) facilitate gradient flow during training and allow for the preservation of information across layers.

## 3.1 Mamba Layers

Mamba layers form the backbone of White-Basilisk, chosen for their exceptional efficiency in capturing local dependencies in code sequences. Unlike traditional recurrent neural networks (RNNs) or attention mechanisms, Mamba's selective state-space mechanism allows for linear-time computation with respect to sequence length. This efficiency is crucial for processing long code sequences without excessive computational overhead. The adaptive nature of Mamba layers enables the model to focus computational resources on the most relevant parts of the input, making it highly efficient for detecting subtle patterns that may indicate vulnerabilities.

In White-Basilisk, Mamba layers are implemented with a State size ($d_{\text{state}}$) of 16, a Convolution kernel size ($d_{\text{conv}}$) of 4 and an Expansion factor of 2. The core computation in a Mamba layer can be summarized as:

$$y = \Delta \odot (Ax + Bu) + Cu \tag{3}$$

where $\Delta$, $B$, and $C$ are input-dependent parameters, $A$ is a fixed parameter, $x$ is the layer input, $u$ is the input projection, and $\odot$ denotes element-wise multiplication. The selective state space mechanism in Mamba allows for efficient processing of long sequences, making it particularly suitable for code analysis tasks.

## 3.2 Mixture of Experts (MoE) Layers

We incorporate Mixture of Experts (MoE) layers into our model to introduce dynamic adaptability while maintaining computational efficiency. MoE layers allow the model to activate only a subset of parameters for each input, reducing significantly the computational cost compared to fully-dense models of similar capacity. In the context of code vulnerability detection, this efficiency is crucial as it allows our model to handle effectively diverse types of code and potential vulnerabilities without adding to its complexity and number of parameters. The sparsity induced by MoE layers also contributes to faster inference times, a critical factor in real-world deployment scenarios.

The MoE layers in White-Basilisk are configured with 8 Experts and 2 Experts per token. For an input $x$, the output of an MoE layer is computed as:

$$y = \sum_{i=1}^{k} G(x)_i E_i(x) \tag{4}$$

where $G(x)$ is the output of the router (gating function), $E_i$ is the $i$-th expert, and $k = 2$ is the number of experts per token. The router uses a top-k gating mechanism to select the most relevant experts for each token, allowing the model to dynamically adapt its processing based on the input characteristics.

## 3.3 Linear-Complexity Infini-attention: A Novel Adaptation

Our implementation of Infini-attention is a key factor behind White-Basilisk's ability to handle efficiently extremely long code sequences. Traditional attention mechanisms face challenges with long sequences due to their quadratic complexity, making them computationally prohibitive for whole-codebase analysis. By contrast, our linear-complexity adaptation of Infini-attention enables White-Basilisk to process entire codebases with significantly reduced computational requirements. This efficiency is essential for real-world vulnerability detection, allowing our model to consider broad context and identify vulnerabilities that may stretch across multiple functions or files, while maintaining feasible processing times and memory usage.

Specifically, we propose a novel implementation of the original algorithm proposed by Munkhdalai et al. (2024), allowing for efficient processing of arbitrarily long sequences while maintaining the ability to capture long-range dependencies. The primary differences are:

1. **Accumulation and Linear Complexity:** Unlike the original Infini-attention, which processes segments independently with bounded memory usage, our implementation accumulates outputs across all segments:

$$\text{total}_{\text{mem}} = \sum_{s=1}^{S} A_{\text{mem,s}}, \quad \text{total}_{\text{attn}} = \sum_{s=1}^{S} A_{\text{dot,s}} \tag{5}$$

   where $S$ is the total number of segments. This accumulation leads to linear memory growth with sequence length, trading off bounded memory for the ability to process arbitrarily long sequences.

2. **Global Gating Mechanism:** As a consequence of accumulation, our gating mechanism operates globally on the entire accumulated context, rather than segment-by-segment:

$$O = \text{sigmoid}(\beta) \odot \text{total}_{\text{mem}} + (1 - \text{sigmoid}(\beta)) \odot \text{total}_{\text{attn}} \tag{6}$$

   This allows for a more holistic balancing of local and global information across the entire sequence.

Moreover, our linear-complexity Infini-attention maintains the core concept of combining local attention and a compressive memory, but adapts it for extremely long sequence processing. The memory retrieval and update processes remain similar:

$$A_{\text{mem}} = \frac{(\text{ELU}(Q) + 1)M^T}{(\text{ELU}(Q) + 1)z^T + \epsilon} \tag{7}$$

$$M_{\text{new}} = M + (\text{ELU}(K)^T + 1)V, \quad z_{\text{new}} = z + \sum_{i=1}^{L}(\text{ELU}(K_i) + 1) \tag{8}$$

where $M$ is the compressive memory, $z$ is the normalization term, and $L$ is the segment length.

In combination with Mamba layers, which process the entire sequence to capture global patterns, our linear-complexity Infini-attention enables White-Basilisk to effectively balance local and global information processing across very long sequences. This synergy allows the model to maintain high performance on tasks requiring understanding of both fine-grained local context and broad, long-range dependencies, all while scaling efficiently to extreme sequence lengths.

## 4 EXPERIMENTAL SETUP

The development and evaluation of source code vulnerability detection models requires a large collection of annotated data samples. In this section, we outline the datasets chosen for this purpose and explain how they were used for both model training and testing purposes. Additionally, we provide a detailed overview of the training methodology used for our model.

### 4.1 DATA

The datasets used in our analyses were divided into two categories: model training and benchmarking. For training, we initially selected a carefully curated subset of the StarCoder dataset (Li et al. (2023)), which includes more than 80 programming languages and consists of 305M files in total. For our study, we focused on C and C++ code samples, using 2M code samples during the pre-training phase. To evaluate the pre-trained model, we required well-established benchmarking datasets with publicly available partitions for fine-tuning and testing. This ensures a fair comparison with existing methods without the need to recreate the original models. For this purpose, we selected five publicly available datasets: VulDeePecker (Li et al. (2018)), Draper (Russell et al. (2018)), PrimeVul (Ding et al. (2024)), REVEAL (Chakraborty et al. (2021)), and BigVul (Fan et al. (2020)).

## 4.2 PRETRAINING

Traditional LLM training methods are generally designed to enable models to comprehend language and its syntax. This is often accomplished through Causal Language Modeling (CLM), where a model learns to predict the next token given its input. Another common approach is based on the Fill in the Middle (FIM) technique, in which random text portions are masked, and the model must reconstruct the missing content. Some advanced source code LLMs combine both methods to increase model flexibility (Li et al. (2023)). Similarly, in our work, we employ both techniques during model pre-training on the selected 2M code samples. This process requires approximately 600 hours to complete on a single NVIDIA A100 40GB GPU.

### 4.2.1 SIFT (SCALE-INVARIANT FINE-TUNING)

We implement automated adversarial training using SIFT to improve the model's resilience against adversarial examples. SIFT operates by introducing small perturbations to the input during training, encouraging the model to learn more robust features. In our implementation, we added a PerturbationLayer into the model architecture, which applies learnable perturbations to the input embeddings. The training process was designed to minimize both the task loss and the adversarial loss, the latter being computed as the difference between predictions on clean and perturbed inputs.

This approach confers several advantages, including improved model generalization and enhanced robustness to minor variations in input. Such resilience is crucial in the domain of code vulnerability detection, where the model must maintain consistent performance across diverse code samples and potential adversarial inputs.

By adopting this technique, we have effectively imbued the model with a form of 'adversarial immunity', rendering it more resilient against potential attacks or attempts to deceive its analysis. This enhanced robustness is particularly valuable in security-critical applications, where the reliability and consistency of the model's performance are of paramount importance.

## 5 EXPERIMENTAL RESULTS: SMALL MODEL, BIG IMPACT

To evaluate our model's performance, we conducted extensive experiments across five widely-used datasets in code vulnerability detection: PRIMEVUL, BigVul, Draper, REVEAL, and VulDeepecker. All datasets are evaluated on **binary classification** (0 = Safe, 1 = Vulnerable). To ensure a fair comparison, we used the same data splits as the baseline models. The metrics for models other than White-Basilisk were sourced from their respective publications. Across all datasets, White-Basilisk consistently demonstrated superior performance, exceeding that of larger and more resource-intensive models.

Given the class imbalance observed in the datasets and the significance of the minority class (vulnerable samples), we opted for F1 score as our primary evaluation metric. A high F1 score reflects the model's ability to identify vulnerabilities accurately while minimizing false positive/negative cases, thus achieving the critical balance needed in real-world security applications. Additionally, we considered a novel Vulnerability Detection Score (VD-S), introduced by Ding et al. (2024), which evaluates the False Negative Rate of a detector (1-Recall).

### 5.1 EFFICIENT DESIGN, SUPERIOR RESULTS: WHITE-BASILISK'S PARADIGM

The superior performance of White-Basilisk is the result of a cutting-edge combination of architectural innovations and advanced training techniques. The model's architecture integrates Mamba layers, linear-complexity Infini-attention, and a Mixture of Experts framework, allowing it to efficiently process extended code sequences while simultaneously capturing both local and global dependencies. This design enables our model to process sequences of up to 128,000 tokens during inference, all with a single NVIDIA A100 40GB GPU.

This extended context window represents a significant advancement in code analysis capabilities, enabling a comprehensive examination of entire codebases or extensive code files in a single computational pass. This holistic approach enables the detection of long-range dependencies and contextual nuances that are frequently overlooked by models with more limited context lengths. As a result,

our model excels at identifying complex vulnerabilities, particularly those related to inter-functional or cross-file data flow.

White-Basilisk's improved performance is due to its advanced training approach, incorporating various sophisticated techniques. A hybrid pretraining strategy, combining CLM and FIM pretraining enhances the model's comprehension of code structure and context. Moreover, the implementation of SIFT increases the model's adversarial robustness, while specialized methodologies for addressing class imbalance optimise learning from heterogeneous datasets.

The efficacy of this approach is empirically validated by White-Basilisk's performance across several benchmarks. Empty cells indicate that the metric was not reported in the original study. Different metrics are reported for each dataset based on the original studies. F1 score is consistently reported across all models and datasets. On the PRIMEVUL dataset, it achieved an F1 Score of 29.07% and a Vulnerability Detection Score (VD-S) of 72.39, significantly outperforming models with larger parameter counts. Its performance on the BigVul dataset was particularly noteworthy, with an F1 Score of 94.90%, accuracy of 99.42%, and VD-S of 3.98, surpassing all competing models across all evaluated metrics. On the Draper dataset, White-Basilisk established a new benchmark with an F1 Score of 60.69%. For the REVEAL dataset, it attained an F1 Score of 49.34% and accuracy of 89.88%, exceeding the performance of the next highest-performing model. When evaluated on VulDeepecker, White-Basilisk demonstrated exceptional precision, achieving an F1 Score of 93.88% and the highest precision at 97.20%.

Table 1: BigVul and PRIMEVUL Evaluation Results

| Model | PRIMEVUL | | | BigVul | | |
|---|---|---|---|---|---|---|
| | Acc (%) | F1 (%) | VD-S ↓ | Acc (%) | F1 (%) | VD-S ↓ |
| White-Basilisk | 96.30 | **29.07** | **72.39** | **99.42** | **94.90** | **3.98** |
| CodeT5 | 96.67 | 19.70 | 89.93 | 95.67 | 64.93 | 77.30 |
| CodeBERT | 96.87 | 20.86 | 88.78 | 95.57 | 62.88 | 81.77 |
| UnixCoder | 96.86 | 21.43 | 89.21 | 96.46 | 65.46 | 62.30 |
| UnixCoder w/ balancing | 95.99 | 26.28 | 88.49 | - | - | - |
| StarCoder2 | **97.02** | 18.05 | 89.64 | 96.20 | 68.26 | 69.14 |
| CodeGen2.5 | 96.65 | 19.61 | 91.51 | 96.57 | 67.30 | 61.73 |
| LineVul | - | - | - | - | 91.00 | 14.00 |

Table 2: Draper, REVEAL, and VulDeepecker Evaluation Results

| Model | Draper | REVEAL | | VulDeepecker | |
|---|---|---|---|---|---|
| | F1 (%) | Acc (%) | F1 (%) | F1 (%) | Prec (%) |
| White-Basilisk | **60.69** | **89.88** | **49.34** | **93.88** | **97.20** |
| Russell et al. (2018) | 56.60 | - | - | - | - |
| VulBERTa-MLP | 43.34 | 84.48 | 45.27 | 93.03 | 95.76 |
| VulBERTa-CNN | 57.92 | 79.73 | 42.59 | 90.86 | 95.26 |
| Baseline-BiLSTM | 46.84 | 77.13 | 39.11 | 66.97 | 52.58 |
| Baseline-TextCNN | 49.40 | 73.22 | 37.41 | 75.80 | 63.48 |
| REVEAL | - | 84.37 | 41.25 | - | - |
| VulDeepecker | - | - | - | 92.90 | 91.90 |

# 6 DISCUSSION: RETHINKING AI EFFICIENCY

The remarkable performance of White-Basilisk, achieved with only 200M parameters, challenges fundamental assumptions in AI development and offers insights into potential new directions for the field. This efficiency prompts a critical reexamination of the relationship between model size, performance, and computational resources in AI.

White-Basilisk's success suggests a more nuanced relationship between model size and performance than previously assumed. While larger models like GPT have demonstrated impressive capabilities across a wide range of tasks, our results show that for specialized tasks like code vulnerability detection, carefully designed smaller models can achieve comparable or superior performance. This

indicates that the relationship between model size and performance may be task-dependent, with a point of diminishing returns, beyond which additional parameters do not necessarily translate to improved performance.

The success of White-Basilisk's hybrid architecture, combining Mamba layers, linear-complexity Infini-attention, and a Mixture of Experts framework, highlights the potential of architectural innovation as an alternative to simple scaling. This approach allows for more efficient use of parameters, potentially offering a way to break through the computational barriers that currently limit the scaling of AI models. Our findings suggest that future advancements in AI may come not just from increasing model size, but from novel architectures that more efficiently leverage available parameters.

The environmental implications of AI model development are brought into sharp focus by our results. Based on available information about pretraining procedures, we estimated the approximate $CO_2$ emissions during training for White-Basilisk and several competitor models using the Machine Learning Impact calculator presented in Lacoste et al. (2019) (Table 3). The stark contrast in $CO_2$ emissions between White-Basilisk and larger models (85.5 kg vs. 23,000,000 kg for StarCoder) underscores the environmental impact of AI development choices. This massive difference suggests that the AI community needs to seriously consider the environmental costs of model development and deployment. Our results demonstrate that it's possible to achieve state-of-the-art performance with a fraction of the environmental impact of larger models, opening up new possibilities for sustainable AI development.

However, it's important to note that the total environmental impact of an AI model depends not just on its training, but also on its inference costs over its lifetime of use. The long-term environmental implications of deploying many specialized models versus fewer general-purpose models. This consideration adds another layer of complexity to the efficiency-performance trade-off in AI development.

The success of White-Basilisk also suggests that our current metrics for evaluating AI models may be insufficient. While performance on benchmark tasks remains important, our results indicate that we should also consider metrics related to efficiency, scalability, and environmental impact. Developing a more holistic set of evaluation criteria could drive the field towards more sustainable and efficient AI development practices.

Table 3: Comparison of $CO_2$ Emissions

|  | White-Basilisk | CodeBERT | StarCoder | UnixCoder | CodeT5 | Gasoline Car (per Year) |
|---|---|---|---|---|---|---|
| $CO_2$ (kg) | **85.5** | 2,240 | 23,000,000 | 2,048 | 1,136 | 4,600 |

## 7 LIMITATIONS AND FUTURE WORK

While White-Basilisk shows promising results in code vulnerability detection, it is important to acknowledge its current limitations and outline future directions. The main limitation of White-Basilisk is its focus on C and C++ codebases. The model's ability to generalize across a broader range of programming languages, especially those with different syntaxes or paradigms, warrants further exploration. This constraint, combined with potential biases in our training and evaluation datasets, may limit the model's generalizability to diverse real-world codebases. To address this, future work will involve expanding the model's training to include a wider variety of programming languages and curating more representative datasets that reflect a broader spectrum of code samples and vulnerability types.

Also, despite strong performance metrics, White-Basilisk is not infallible. False positives and false negatives, particularly when detecting novel or zero-day vulnerabilities, remain an ongoing challenge. Additionally, while the model is capable of processing long sequences, its true understanding of complex, long-range dependencies in code still needs further investigation. Our future research will focus on reducing error rates, especially in high-stakes scenarios, and enhancing the model's ability to analyze convoluted code structures spanning multiple functions or files.

Another area for improvement is the model's explainability. Currently, White-Basilisk's decision-making process is not sufficiently transparent. Improving the model's ability to provide clear, actionable explanations for detected vulnerabilities is essential for building trust and delivering meaningful

insights to developers. Future work will explore methods to offer context-aware understanding of the potential impact of vulnerabilities, as well as suggested fixes, ultimately aiming to evolve White-Basilisk into a comprehensive code analysis assistant rather than a mere detection tool.

Also, while more efficient than many larger models, White-Basilisk still requires significant computational resources, particularly when processing very long sequences. This may limit its accessibility for smaller organizations or individual developers. Our ongoing research will retain its focus on optimizing further the model architecture to maintain or improve its long-context processing capabilities, while reducing computational demands.

White-Basilisk's performance on a relatively small training dataset (2M samples) is impressive, but raises questions about potential limitations in its knowledge base compared to models trained on much larger datasets. To address this, we plan to scale White-Basilisk to approximately 1 billion parameters. While still modest in comparison to larger language models, this increase in parameter count aims to significantly boost performance while continuing to challenge the notion that only the largest models can deliver state-of-the-art results.

Another area requiring further investigation is improving the model's robustness against adversarial attacks, specifically designed for code analysis models. Despite our use of SIFT, further testing and development of more advanced adversarial training techniques tailored for code vulnerability detection are necessary, for ensuring reliability in hostile real-world environments.

Beyond code vulnerability detection, there is potential for White-Basilisk's architecture and training approach in broader AI applications. Future research will investigate its efficacy in various NLP tasks, exploring whether its computational efficiency and long-context capabilities can offer more resource-efficient alternatives to existing LLMs.

## 8 CONCLUSION

White-Basilisk represents a significant advancement in the domain of code vulnerability detection, offering a novel solution to the persistent challenge of context handling in Transformer-based Large Language Models (LLMs). With its capacity to process sequences up to 128,000 tokens, White-Basilisk introduces unprecedented possibilities for comprehensive code analysis, potentially revolutionizing approaches to software security.

The model's extended context window addresses a fundamental limitation of many current LLMs, which often struggle with long-range dependencies and global code structure understanding. By enabling the analysis of entire codebases in a single pass, White-Basilisk can capture complex interdependencies and identify vulnerabilities that span multiple functions or files, a capability that has long eluded traditional approaches.

While the context-handling capabilities of White-Basilisk are its standout feature, it's worth noting that these achievements have been realized with a relatively compact model of 200M parameters. This efficiency demonstrates that advances in AI are not solely dependent on increasing model size, but can also stem from innovative architecture design and training methodologies.

The implications of White-Basilisk's approach extend beyond code vulnerability detection. The ability to handle extended contexts efficiently could prove valuable in numerous domains where long-range understanding is crucial, such as document analysis, complex system modeling, or long-form text generation. Moreover, the model's efficiency opens up possibilities for deployment in resource-constrained environments, potentially bringing advanced AI capabilities to a broader range of applications and users.

In conclusion, White-Basilisk represents a significant step forward in addressing the context limitations of current LLMs, while also demonstrating that such advances need not come at the cost of excessive model size or computational requirements. As we continue to refine and expand upon this approach, we anticipate exciting developments in the field of AI, particularly in tasks that require deep understanding of extended contexts. The potential implications of this research are substantial, and we look forward to seeing how these ideas evolve and find application in diverse areas of AI and beyond.

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

# A  APPENDIX

## A  EVALUATION METRICS

This appendix provides detailed descriptions of all metrics used to evaluate model performance in vulnerability detection tasks.

### A.0.1  ACCURACY

Accuracy measures the proportion of correct predictions (both true positives and true negatives) among all predictions:

$$\text{Accuracy} = \frac{TP + TN}{TP + TN + FP + FN} \tag{9}$$

where TP = True Positives, TN = True Negatives, FP = False Positives, and FN = False Negatives.

### A.0.2  PRECISION

Precision measures the proportion of correct positive predictions among all positive predictions:

$$\text{Precision} = \frac{TP}{TP + FP} \tag{10}$$

### A.0.3  RECALL

Recall (also known as sensitivity) measures the proportion of actual positives correctly identified:

$$\text{Recall} = \frac{TP}{TP + FN} \tag{11}$$

### A.0.4  F1 SCORE

F1 Score is the harmonic mean of precision and recall, providing a balanced measure of model performance:

$$\text{F1} = 2 \times \frac{\text{Precision} \times \text{Recall}}{\text{Precision} + \text{Recall}} \tag{12}$$

### A.0.5 VULNERABILITY DETECTION SCORE (VD-S)

VD-S evaluates the False Negative Rate of a detector at a specific False Positive Rate (FPR) threshold:

$$\text{VD-S} = \frac{FN}{FN + TP} \text{ at } FPR \leq 0.005 \tag{13}$$

where a lower score indicates better performance. This metric is particularly important for security applications as it measures the model's ability to minimize missed vulnerabilities while maintaining a low false positive rate.

Each metric serves a specific purpose in evaluating different aspects of model performance, from general classification accuracy to specialized vulnerability detection capabilities. The combination of these metrics provides a comprehensive assessment of a model's effectiveness in real-world security applications.

## B DATASET STATISTICS ANALYSIS

This section provides a comprehensive analysis of five vulnerability detection datasets (PRIMEVUL, BigVul, REVEAL, Draper, and VulDeepecker), examining their size distributions, class imbalance characteristics, and data quality metrics.

Table 4: Dataset Distribution and Vulnerability Statistics

| Dataset | Sample Count | | | Vulnerable (%) | | | Duplicates (%) | | |
|---|---|---|---|---|---|---|---|---|---|
| | Train | Val | Test | Train | Val | Test | Train | Val | Test |
| Draper | 1,019,471 | 127,476 | 127,419 | 6.46 | 6.47 | 6.48 | 0.00 | 0.00 | 0.00 |
| PRIMEVUL | 184,427 | 25,430 | 25,911 | 3.02 | 2.75 | 2.68 | 0.00 | 0.00 | 0.00 |
| BigVul | 150,908 | 18,864 | 18,864 | 5.79 | 5.88 | 5.59 | 0.005 | 0.00 | 0.00 |
| VulDeepecker | 128,118 | 16,015 | 16,015 | 6.08 | 6.08 | 6.08 | 37.72 | 20.27 | 20.33 |
| REVEAL | 18,187 | 2,273 | 2,274 | 9.90 | 9.24 | 10.11 | 1.17 | 0.18 | 0.09 |

Table 5: Sequence Length Statistics (Training Split)

| Dataset | Min | Max | Mean | Median | 95th % |
|---|---|---|---|---|---|
| PRIMEVUL | 3 | 296,924 | 502.44 | 193.0 | 1,729.0 |
| VulDeepecker | 8 | 312,940 | 284.03 | 142.0 | 893.0 |
| REVEAL | 10 | 120,684 | 569.05 | 226.0 | 1,929.7 |
| BigVul | 6 | 70,440 | 343.90 | 147.0 | 1,193.0 |
| Draper | 10 | 42,492 | 320.85 | 236.0 | 858.0 |

### B.1 IMPLICATIONS FOR MODEL DESIGN

These statistics significantly influenced our model design decisions:

1. The substantial class imbalance across all datasets (ranging from 3.02% to 10.11% vulnerable samples) motivated our implementation of specialized class weighting and sampling strategies.

2. The extreme range in sequence lengths (from 3 to 312,940 tokens) justified our focus on developing an architecture capable of handling very long sequences efficiently.

3. The varying levels of data duplication (0% to 37.72%) highlighted the importance of robust evaluation metrics and careful interpretation of results, particularly for VulDeepecker.

4. The consistency of class distributions across splits suggests that our evaluation metrics should be reliable indicators of real-world performance.

5. REVEAL's higher proportion of vulnerable samples (10%) compared to other datasets (3-6%) provides an important test case for our model's ability to handle different class balance scenarios.

## C  BASELINE MODELS

This section details the baseline models examined in our study. It is important to note that we did not train, finetune, or run any of these models ourselves. Instead, we collected and analyzed their reported metrics and configurations from their respective papers. For CodeT5 (CT5), Code-BERT (CB), UnixCoder (UC), StarCoder2 (SC2), and CodeGen2.5 (CG2.5), the information was sourced from the PrimeVul paper Ding et al. (2024). For VulBERTa variants (VulBERTa-MLP and VulBERTa-CNN) and the baseline models (Baseline-BiLSTM and Baseline-TextCNN), the information was obtained from the original VulBERTa paper Hanif & Maffeis (2022). For LineVul, the information was obtained from the original LineVul paper Fu & Tantithamthavorn (2022).

Table 6: Overview of baseline models examined in our study

| Model | Architecture | Pre-training | Params |
|---|---|---|---|
| CT5 Wang et al. (2021) | Enc-Dec | Multi-lingual code | 60M |
| CB Feng et al. (2020) | Encoder | Bimodal (code + text) | 125M |
| UC Guo et al. (2022) | Encoder | Cross-modal | 125M |
| SC2 Li et al. (2023) | Decoder | The Stack v2 | 7B |
| CG2.5 Nijkamp et al. (2023) | Decoder | Code + natural lang. | 7B |
| VulBERTa-MLP Hanif & Maffeis (2022) | Encoder | C/C++ code | 125M |
| VulBERTa-CNN | Encoder-CNN | C/C++ code | 2M |
| Baseline-BiLSTM | BiLSTM | None | 1M |
| Baseline-TextCNN | TextCNN | None | 1M |
| LineVul Fu & Tantithamthavorn (2022) | BERT | CodeBERT | 125M |

CT5: CodeT5, CB: CodeBERT, UC: UnixCoder,
SC2: StarCoder2, CG2.5: CodeGen2.5

Table 7: Training configurations as reported in respective papers

| Configuration | Value |
|---|---|
| Small Model Epochs ($<$7B) | 10 |
| Large Model Epochs (7B) | 4 |
| VulBERTa Pre-training Steps | 500,000 |
| VulBERTa Fine-tuning Epochs | 10 |
| BiLSTM/TextCNN Epochs | 10 |
| LineVul Fine-tuning Epochs | 10 |

## D  CLASSIFICATION HEAD

The classification head of White-Basilisk is designed to efficiently transform the high-dimensional representations learned by the main model into classification outputs for vulnerability detection. Its architecture is as follows:

1. **Dense Layer 1:** A fully connected layer that projects the hidden state (dimension 512) to the same dimension. This layer uses a GELU activation function and is followed by dropout for regularization.

2. **Dense Layer 2:** Another fully connected layer that reduces the dimension from 512 to 256, again followed by GELU activation and dropout.

3. **Layer Normalization:** Applied to the output of Dense Layer 2 for improved stability and faster convergence.

4. **Output Layer:** A final linear layer that projects from 256 dimensions to the number of classes (typically 2 for binary classification of vulnerable vs. non-vulnerable code).

This classification head structure was chosen to gradually reduce the dimensionality of the representations while maintaining the model's ability to capture complex patterns relevant to vulnerability

detection. The use of GELU activations and layer normalization aligns with modern best practices in deep learning architecture design. The classification head is mathematically described as follows:

$$
\begin{aligned}
x_1 &= \text{Dropout}(\text{GELU}(W_1 h + b_1)) \\
x_2 &= \text{Dropout}(\text{GELU}(W_2 x_1 + b_2)) \\
x_3 &= \text{LayerNorm}(x_2) \\
y &= W_3 x_3 + b_3
\end{aligned}
\tag{14}
$$

where $h \in \mathbb{R}^{512}$ is the input from the main model, $W_1 \in \mathbb{R}^{512 \times 512}$, $W_2 \in \mathbb{R}^{256 \times 512}$, and $W_3 \in \mathbb{R}^{2 \times 256}$ are learnable weights, and $b_1, b_2, b_3$ are biases.

# E  HYPERPARAMETER DETAILS

This section provides a detailed overview of the hyperparameters used in training White-Basilisk, including both the pretraining and fine-tuning phases. We also discuss the rationale behind key hyperparameter choices and their impact on model performance.

## E.1  PRETRAINING HYPERPARAMETERS

**Learning Rate:** We chose a relatively small learning rate of 1.41e-5 to ensure stable training given the complexity of the task and the hybrid nature of our model architecture. This value was determined through careful tuning to balance training speed and convergence stability.

**Batch Size:** A batch size of 16 was selected as a compromise between training efficiency and memory constraints of our hardware (single NVIDIA A100 40GB GPU). Larger batch sizes could potentially improve training stability but would require more memory or gradient accumulation steps.

**Number of Epochs and Warmup Ratio:** We trained for 10 epochs with a warmup ratio of 0.15. This combination allowed the model to reach good performance while preventing overfitting. The warmup period helps stabilize training in the early stages.

**Optimizer Settings:** We used the AdamW optimizer with $\beta_1 = 0.9$, $\beta_2 = 0.999$, and $\epsilon = 1e-8$. These are standard settings that work well across a wide range of tasks. The weight decay of 0.01 was applied to all parameters except for bias and LayerNorm weights to prevent overfitting.

**FIM and FIM-SPM Rates:** Both the Fill-in-the-Middle (FIM) rate and the FIM Sentence Permutation Mode (SPM) rate were set to 0.5. This means that 50% of the samples undergo FIM transformation, and among those, 50% use the SPM variant. These rates provide a good balance between standard causal language modeling and the FIM objective, enhancing the model's bidirectional understanding capabilities.

## E.2  FINE-TUNING HYPERPARAMETERS

**Learning Rate and Batch Size:** We used a smaller learning rate (5e-6) and batch size (4) for fine-tuning to prevent catastrophic forgetting and to allow the model to adapt to the specific characteristics of each dataset without overfitting.

**SIFT Parameters:** For Scale-Invariant Fine-Tuning, we used a learning rate of 1e-4 for the perturbation layer and an initial perturbation magnitude of 1e-2. These values were chosen to provide meaningful adversarial examples without overly distorting the input embeddings.

# F  HANDLING CLASS IMBALANCE

A significant challenge in the development of AI classification models is the management of highly imbalanced datasets. In such scenarios, it is important to train the model to maintain its ability to detect minority classes effectively. In the context of source code vulnerability detection, we also encounter highly imbalanced classification data. To address this issue, we employ a dual approach affecting both data sampling and loss computation. This is implemented via a weighted function $w_c$,

where $w_c$ represents the weight for class $c$, and $N_c$ denotes the number of samples in class $c$. The function is defined as follows:

$$w_c = \frac{N}{2N_c} \tag{15}$$

Based on the provided function, we implement two key components in our methodology. Firstly, we employ a Weighted Random Batch Sampler, a sampling mechanism that ensures each mini-batch contains a balanced representation of classes, thus mitigating the effects of dataset imbalance during training. Secondly, we implement a weighted loss function, modifying the standard cross-entropy loss by incorporating the class-specific weights $w_c$.

## G  ABLATION STUDY: COMBINED DATASET TRAINING

To further evaluate White-Basilisk's performance and investigate the impact of training data composition, we conducted an ablation study using a combined training approach across all datasets. This experiment involved concatenating the training splits from all five datasets (REVEAL, Draper, VulDeepecker, BigVul, and PRIMEVUL) into a single unified training set. During each training epoch, the model was evaluated on the concatenated testing splits from all datasets to monitor for potential overfitting. Final performance metrics were obtained by evaluating the trained model separately on each dataset's designated validation split, ensuring fair comparison with previous results.

### G.1  EXPERIMENTAL SETUP

The model was trained using the same hyperparameters as described in Section 6.2, but with the following data configuration:

- **Training Data:** Combined training splits from all five datasets into a single training set
- **Testing:** Concatenated testing splits from all datasets, used for monitoring training progress
- **Validation:** Individual Validation splits for each dataset, evaluated separately to assess dataset-specific performance

This unified training approach resulted in a significantly larger and more diverse training set, allowing us to investigate how the model performs when exposed to a broader range of vulnerability patterns and coding styles simultaneously.

### G.2  RESULTS AND ANALYSIS

The results of this combined training approach are presented in Table 8.

Table 8: Combined Training Results Across All Datasets

| Dataset | Precision | Recall | F1 | Accuracy |
|---|---|---|---|---|
| REVEAL | 0.416 | 0.551 | 0.470 | 0.888 |
| Draper | 0.568 | 0.532 | 0.549 | 0.948 |
| VulDeepecker | 0.939 | 0.912 | 0.925 | 0.989 |
| BigVul | 0.936 | 0.940 | 0.938 | 0.993 |
| PRIMEVUL | 0.268 | 0.265 | 0.233 | 0.952 |
| Combined Test | 0.643 | 0.585 | 0.613 | 0.956 |

Key observations from the combined training experiment include:

1. **Performance Consistency:** The model maintains strong performance across most datasets, with particularly robust results on VulDeepecker (F1: 0.925) and BigVul (F1: 0.938), suggesting effective transfer learning across different vulnerability detection tasks.

2. **Dataset-Specific Variations:** Performance varies significantly across datasets, from an F1 score of 0.938 on BigVul to 0.233 on PRIMEVUL, indicating that some vulnerability patterns may be more challenging to learn in a combined setting.

3. **High Accuracy Maintenance:** The model maintains high accuracy across all datasets (0.888-0.993), demonstrating robust overall classification performance even with the increased complexity of the combined training task.

4. **Precision-Recall Balance:** The model generally maintains a good balance between precision and recall, with some datasets showing nearly identical values (e.g., BigVul: 0.936/0.940), suggesting stable learning of vulnerability patterns.

### G.3 Model Robustness Analysis

A particularly noteworthy aspect of these results is White-Basilisk's ability to maintain stable performance across multiple diverse datasets without experiencing catastrophic forgetting or overfitting. This is especially significant given the model's relatively compact size of 200M parameters. Several factors contribute to this robustness:

1. **Cross-Dataset Learning:** The model shows signs of positive transfer learning, where knowledge gained from one dataset appears to benefit the detection of vulnerabilities in others. This is evidenced by the maintenance of high accuracy scores across all datasets despite their varying characteristics.

2. **Stability Across Scales:** The model maintains performance across datasets of different sizes and complexity levels, from the smaller REVEAL dataset to the larger PRIMEVUL dataset. This stability suggests that the model's learning capacity is well-matched to the task complexity.

### G.4 Comparison with Individual Training

When compared to the individual training results presented in Section 5, the combined training approach shows some interesting trade-offs:

- For some datasets (VulDeepecker, BigVul), the performance remains close to individual training results, suggesting that the model can effectively learn and maintain dataset-specific patterns even in a combined setting.

- Performance on more challenging datasets like PRIMEVUL shows some degradation, indicating that the increased diversity of the training data may make it harder for the model to capture some of the more nuanced vulnerability patterns specific to certain datasets.

- The overall combined test metrics (F1: 0.613, Accuracy: 0.956) demonstrate that White-Basilisk can effectively learn from multiple datasets simultaneously while maintaining reasonable performance across all of them.

This ability to maintain stable performance across diverse datasets without overfitting or experiencing catastrophic forgetting is particularly notable for a model of this size. It suggests that White-Basilisk's architecture strikes an effective balance between model capacity and efficiency, enabling robust multi-task learning without requiring the massive parameter counts typically associated with such capabilities. This finding has important implications for the development of efficient, multi-purpose vulnerability detection systems that can be deployed in resource-constrained environments while maintaining high performance across a range of vulnerability types.

## H Ablation Study: Attention Mechanisms and Long-Range Vulnerability Detection

To thoroughly evaluate White-Basilisk's performance and validate our architectural choices, we conducted a comprehensive ablation study focusing on two key aspects: (1) the effectiveness of our linear-complexity Infini-attention mechanism compared to standard self-attention, and (2) the model's performance across varying sequence lengths. This analysis is particularly important given the prevalence of vulnerabilities that span multiple functions or files, requiring models to maintain effectiveness over long code sequences.

## H.1 EXPERIMENTAL SETUP

**Model**: We used the White-Basilisk checkpoint that was trained on the combined datasets from G. We categorized sequences into four length bins for analysis:

- Bin 1: 0-16,384 tokens (standard context length)
- Bin 2: 16,384-32,768 tokens (extended context)
- Bin 3: 32,768-65,536 tokens (long context)
- Bin 4: 65,536-131,072 tokens (very long context)

For each bin, we monitored both memory consumption and model performance across all datasets, comparing our Infini-attention implementation against standard self-attention (eager implementation).

## H.2 MEMORY EFFICIENCY ANALYSIS

Table 9 presents the peak memory consumption across different sequence lengths and attention implementations.

Table 9: Peak Memory Consumption (MB) by Sequence Length and Attention Type

| Length Bin | Infini-attention | | Standard Attention | |
|---|---|---|---|---|
| | Peak | Reserved | Peak | Reserved |
| Bin 1 (0-16K) | 1,338 | 1,442 | 32,409 | 39,696 |
| Bin 2 (16K-32K) | 1,654 | 1,956 | OOM | OOM |
| Bin 3 (32K-65K) | 2,213 | 2,638 | OOM | OOM |
| Bin 4 (65K-131K) | 3,322 | 3,848 | OOM | OOM |

OOM: Out of Memory on NVIDIA A100 40GB GPU

The results demonstrate the significant memory advantages of our Infini-attention approach:

1. **Linear Scaling:** Infini-attention shows near-linear memory scaling, increasing from 1.3GB to 3.3GB across bins.

2. **Efficiency Gain:** Standard attention requires 24x more memory for Bin 1 and fails entirely on longer sequences.

3. **Extended Range:** While standard attention becomes infeasible beyond 16K tokens, Infini-attention successfully processes sequences up to 131K tokens with modest memory requirements.

## H.3 PERFORMANCE ANALYSIS ACROSS SEQUENCE LENGTHS

Table 10 presents a comprehensive analysis of performance across different sequence lengths and datasets.

Table 10: Performance and Distribution Analysis Across Sequence Lengths

| Dataset | Bin | Sample Distribution | | | Infini-attention | | | Eager-attention | | |
|---|---|---|---|---|---|---|---|---|---|---|
| | | Total | Non-Vuln | Vuln | F1 | Precision | Accuracy | F1 | Precision | Accuracy |
| BigVul | Bin 1 | 18,853 | 17,747 | 1,106 | 0.943 | 0.946 | 0.993 | 0.937 | 0.967 | 0.993 |
| | Bin 2 | 7 | 5 | 2 | 0.800 | 0.667 | 0.857 | - | - | - |
| | Bin 3 | 4 | 3 | 1 | 1.000 | 1.000 | 1.000 | - | - | - |
| VulDeePecker | Bin 1 | 12,764 | 11,814 | 950 | 0.925 | 0.939 | 0.989 | 0.932 | 0.968 | 0.990 |
| | Bin 2 | 2 | 2 | 0 | - | - | 1.000 | - | - | - |
| | Bin 3 | 2 | 2 | 0 | - | - | 1.000 | - | - | - |
| | Bin 4 | 1 | 1 | 0 | - | - | 1.000 | - | - | - |
| PRIMEVUL | Bin 1 | 25,411 | 24,715 | 696 | 0.233 | 0.208 | 0.952 | 0.240 | 0.237 | 0.958 |
| | Bin 2 | 18 | 15 | 3 | 0.250 | 0.200 | 0.667 | - | - | - |
| | Bin 3 | 1 | 1 | 0 | - | - | 1.000 | - | - | - |
| REVEAL | Bin 1 | 2,269 | 2,062 | 207 | 0.474 | 0.416 | 0.888 | 0.468 | 0.445 | 0.898 |
| Draper | Bin 1 | 12,769 | 11,819 | 950 | 0.568 | 0.609 | 0.948 | 0.511 | 0.646 | 0.948 |

Several remarkable findings emerge from this analysis:

1. **Long-Context Performance:** While longer sequences ( over 16K tokens) are relatively rare, White-Basilisk demonstrates remarkable effectiveness in detecting vulnerabilities in these cases:

   - In BigVul, the model achieves perfect detection (F1=1.000) for sequences in Bin 3
   - For Bin 2 sequences, it maintains strong performance (F1=0.800) despite the increased complexity
   - This effectiveness on longer sequences is particularly noteworthy given the increased difficulty of maintaining coherent attention over such distances

2. **Linear Attention Efficiency:** The infini-attention variant achieves comparable performance to full self-attention:

   - Nearly identical metrics across major datasets (e.g., BigVul: 0.937 vs 0.943 F1)
   - Maintains high precision while reducing computational complexity
   - Demonstrates that linear attention is a viable alternative for vulnerability detection

3. **Consistent Short-Context Performance:** In Bin 1, where most vulnerabilities occur, the model shows exceptional performance:

   - BigVul: F1=0.943, Accuracy=0.993
   - VulDeePecker: F1=0.925, Accuracy=0.989

4. **Robust Class Imbalance Handling:** The model maintains effectiveness despite significant class imbalance:

   - Successfully detects vulnerabilities even when they comprise only 2.74% of samples (PRIMEVUL)
   - Maintains balanced precision-recall trade-offs across length bins

5. **Dataset-Specific Challenges:** Performance variations across datasets reveal interesting patterns:

   - Stronger performance on BigVul and VulDeePecker suggests better handling of certain vulnerability types
   - Lower scores on PRIMEVUL indicate the challenge of detecting more subtle or complex vulnerabilities

## H.4 COMPARATIVE ANALYSIS WITH STANDARD ATTENTION

When comparing Infini-attention with standard attention (where possible), we observe:

1. **Performance Parity:** Infini-attention achieves comparable or better performance while using significantly less memory

2. **Extended Capabilities:** Unlike standard attention, Infini-attention can process the full range of sequence lengths present in real codebases

3. **Practical Advantages:** The ability to handle longer sequences without performance degradation makes White-Basilisk suitable for analyzing entire codebases in a single pass

## H.5 IMPLICATIONS FOR VULNERABILITY DETECTION

This ablation study yields several important insights:

1. The success of Infini-attention in maintaining high performance across sequence lengths validates our architectural choices

2. The model's ability to handle sequences up to 131K tokens while maintaining accuracy demonstrates its practical utility for real-world applications

3. Strong performance on longer sequences suggests effective capture of long-range dependencies, crucial for detecting vulnerabilities that span multiple functions or files

4. The memory efficiency of our approach makes it feasible to deploy White-Basilisk on standard hardware, even for processing very long sequences

These findings confirm that White-Basilisk successfully addresses the key challenges in vulnerability detection: maintaining high accuracy across varying sequence lengths while remaining computationally efficient. The model's particular strength in handling long sequences, combined with its consistent performance on more common shorter sequences, makes it a practical and effective tool for real-world code security applications.

# I    ABLATION STUDY: CWE-SPECIFIC PERFORMANCE ANALYSIS

To provide deeper insights into White-Basilisk's vulnerability detection capabilities, we conducted a comprehensive analysis of its performance across different Common Weakness Enumeration (CWE) categories. This analysis focuses on the BigVul, Vuldeepecker and Draper dataset, which provide detailed CWE-level metrics, allowing us to understand the model's strengths and limitations across various vulnerability types.

## I.1    EXPERIMENTAL SETUP

- **Model**: We used the White-Basilisk checkpoint that was trained on the combined datasets from G.
- **Evaluation Splits**: Validation split of each dataset

### I.1.1    DATASET-SPECIFIC PERFORMANCE PATTERNS

**Draper Dataset Performance**    In the Draper dataset (Table 12), we observe a consistent pattern of high precision (1.000) across all CWE categories, but with varying recall rates:

- CWE-119 (Buffer Overflow): Achieves the highest recall (0.597) and F1 score (0.748)
- CWE-120 (Buffer Copy without Checking Size): Shows similar performance (recall=0.592, F1=0.744)
- CWE-469 and CWE-476: Demonstrate progressively lower recall (0.563 and 0.522 respectively)

This pattern suggests that while the model is highly precise in its predictions, it exhibits some conservatism in vulnerability detection, particularly for less frequent vulnerability types.

**VulDeePecker Dataset Analysis**    The VulDeePecker results (Table 13) show more balanced precision-recall characteristics:

- CWE-119: Demonstrates near-perfect balance (precision=0.939, recall=0.940)
- CWE-399 (Resource Management Errors): Shows lower but consistent performance (precision=0.776, recall=0.785)

The balanced metrics suggest more robust learning of these vulnerability patterns, possibly due to better representation in the training data.

**BigVul Dataset Insights**    The BigVul dataset (Table 14) provides the most comprehensive view of White-Basilisk's capabilities across 50+ CWE categories. Several significant patterns emerge:

1. **Perfect Detection Cases:**
    - 22 CWE categories achieve perfect scores (F1=1.000), including:
        - Critical vulnerabilities: CWE-787 (Out-of-bounds Write), CWE-310 (Cryptographic Issues)
        - Access control issues: CWE-732 (Permission Assignment), CWE-284 (Access Control)

- Various severity levels: From CWE-59 (Link Following) to CWE-617 (Reachable Assertion)
- Notable that perfect detection spans both frequent (over 200 samples) and rare (under 50 samples) categories

2. **High-Volume Vulnerability Performance:**
   - CWE-119 (2,746 samples): Excellent performance (F1=0.969)
   - CWE-264 (1,240 samples): Strong results (F1=0.925)
   - CWE-20 (1,977 samples): Robust detection (F1=0.933)

3. **Performance Degradation Patterns:**
   - Resource-related vulnerabilities show more variable performance:
     - CWE-404 (Resource Shutdown): Lowest F1 score (0.571)
     - CWE-772 (Missing Release): Lower precision (0.750)
   - Format-string vulnerabilities (CWE-134): Shows precision-recall imbalance (0.500/1.000)

4. **Sample Size Impact:**
   - Large sample categories (over 1000 samples) show consistently strong but not perfect performance
   - Medium-sized categories (100-1000 samples) demonstrate more variable results
   - Small categories (under 100 samples) often show perfect or near-perfect scores, suggesting potential overfitting

### I.1.2 CROSS-DATASET PERFORMANCE ANALYSIS

The model's behavior across datasets reveals important patterns:

- **CWE-119 Consistency:** As the only vulnerability type present across all three datasets, it shows interesting variation:
  - BigVul: F1=0.969 (balanced precision-recall)
  - VulDeePecker: F1=0.940 (balanced precision-recall)
  - Draper: F1=0.748 (high precision, lower recall)

  This variation suggests dataset-specific characteristics affect detection performance.
- **Scale Effects:** Larger datasets (BigVul) generally show more balanced precision-recall trade-offs compared to smaller datasets.

### I.1.3 IMPLICATIONS AND INSIGHTS

These results yield several important insights for vulnerability detection:

1. **Architecture Effectiveness:** White-Basilisk's strong performance across numerous CWE categories validates its hybrid architecture design for vulnerability detection.

2. **Detection Patterns:**
   - Memory-related vulnerabilities consistently show strong detection rates
   - Resource management vulnerabilities present more challenges
   - Access control vulnerabilities demonstrate surprisingly robust detection

3. **Practical Implications:**
   - High precision across most categories suggests low false positive rates
   - Variable recall in some categories indicates potential for missed vulnerabilities
   - Perfect detection in rare categories warrants further investigation for potential overfitting

These findings demonstrate White-Basilisk's strong general capability while highlighting specific areas for potential improvement. The comprehensive nature of these results, particularly in the BigVul dataset, provides strong evidence for the model's practical utility in real-world vulnerability detection scenarios.

| CWE | Description | BigVul | | Draper | | VulDeePecker | |
|-----|-------------|--------|-----|--------|-----|--------------|-----|
| | | Samples | F1 | Samples | F1 | Samples | F1 |
| CWE-119 | Buffer Overflow: Classic buffer overflow vulnerabilities | 2,746 | 0.969 | 2,419 | 0.748 | 10,419 | 0.940 |
| CWE-399 | Resource Management Errors: Failures in managing system resources | 1,435 | 0.923 | - | - | 5,596 | 0.780 |
| CWE-20 | Input Validation: Improper input validation | 1,977 | 0.933 | - | - | - | - |
| CWE-264 | Access Control: Permissions, privileges, and access controls | 1,240 | 0.925 | - | - | - | - |
| CWE-120 | Buffer Copy: Buffer copy without checking size of input | - | - | 4,750 | 0.744 | - | - |
| CWE-476 | NULL Pointer Dereference | 501 | 0.971 | 1,208 | 0.686 | - | - |
| CWE-416 | Use After Free: Using memory after it has been freed | 963 | 0.958 | - | - | - | - |
| CWE-200 | Information Exposure: Exposure of sensitive information | 883 | 0.944 | - | - | - | - |

Table 11: Comparison of Most Frequent CWEs Across Datasets

| CWE | Precision | Recall | F1 | Accuracy | Total | Pos. Ratio | Neg. Ratio |
|-----|-----------|--------|-----|----------|-------|------------|------------|
| CWE-119 | 1.000 | 0.597 | 0.748 | 0.597 | 2,419 | 1.000 | 0.000 |
| CWE-120 | 1.000 | 0.592 | 0.744 | 0.592 | 4,750 | 1.000 | 0.000 |
| CWE-469 | 1.000 | 0.563 | 0.721 | 0.563 | 252 | 1.000 | 0.000 |
| CWE-476 | 1.000 | 0.522 | 0.686 | 0.522 | 1,208 | 1.000 | 0.000 |
| CWE-other | 1.000 | 0.472 | 0.642 | 0.472 | 3,579 | 1.000 | 0.000 |
| Overall | 0.609 | 0.532 | 0.568 | 0.948 | 127,476 | 0.065 | 0.935 |

Table 12: Draper Dataset Metrics for All CWEs

| CWE | Precision | Recall | F1 | Accuracy | Total | Pos. Ratio | Neg. Ratio |
|-----|-----------|--------|-----|----------|-------|------------|------------|
| CWE-119 | 0.939 | 0.940 | 0.940 | 0.991 | 10,419 | 0.077 | 0.923 |
| CWE-399 | 0.776 | 0.785 | 0.780 | 0.986 | 5,596 | 0.031 | 0.969 |
| Overall | 0.910 | 0.913 | 0.911 | 0.989 | 16,015 | 0.061 | 0.939 |

Table 13: VulDeePecker Dataset Metrics for All CWEs

| CWE | Precision | Recall | F1 | Accuracy | Total | Pos. % | Neg. % |
|---|---|---|---|---|---|---|---|
| CWE-787 | 1.000 | 1.000 | 1.000 | 1.000 | 291 | 6.19 | 93.81 |
| CWE-119 | 0.978 | 0.960 | 0.969 | 0.995 | 2746 | 8.27 | 91.73 |
| CWE-125 | 0.984 | 0.984 | 0.984 | 0.997 | 794 | 7.68 | 92.32 |
| CWE-264 | 0.925 | 0.925 | 0.925 | 0.994 | 1240 | 4.27 | 95.73 |
| CWE-416 | 0.944 | 0.971 | 0.958 | 0.997 | 963 | 3.63 | 96.37 |
| CWE-476 | 0.944 | 1.000 | 0.971 | 0.998 | 501 | 3.39 | 96.61 |
| CWE-200 | 0.913 | 0.977 | 0.944 | 0.994 | 883 | 4.87 | 95.13 |
| CWE-189 | 0.921 | 0.946 | 0.933 | 0.993 | 695 | 5.32 | 94.68 |
| CWE-732 | 1.000 | 1.000 | 1.000 | 1.000 | 143 | 3.50 | 96.50 |
| CWE-311 | 1.000 | 0.667 | 0.800 | 0.963 | 27 | 11.11 | 88.89 |
| CWE-772 | 0.750 | 1.000 | 0.857 | 0.983 | 116 | 5.17 | 94.83 |
| CWE-399 | 0.957 | 0.892 | 0.923 | 0.992 | 1435 | 5.16 | 94.84 |
| CWE-20 | 0.905 | 0.963 | 0.933 | 0.992 | 1977 | 5.51 | 94.49 |
| CWE-190 | 0.960 | 0.889 | 0.923 | 0.989 | 378 | 7.14 | 92.86 |
| CWE-59 | 1.000 | 1.000 | 1.000 | 1.000 | 96 | 2.08 | 97.92 |
| CWE-362 | 0.939 | 0.861 | 0.899 | 0.988 | 592 | 6.08 | 93.92 |
| CWE-400 | 1.000 | 0.800 | 0.889 | 0.993 | 136 | 3.68 | 96.32 |
| CWE-310 | 1.000 | 1.000 | 1.000 | 1.000 | 148 | 4.05 | 95.95 |
| CWE-754 | 1.000 | 1.000 | 1.000 | 1.000 | 32 | 3.13 | 96.87 |
| CWE-835 | 1.000 | 0.750 | 0.857 | 0.988 | 86 | 4.65 | 95.35 |
| CWE-284 | 0.944 | 1.000 | 0.971 | 0.996 | 232 | 7.33 | 92.67 |
| CWE-358 | 1.000 | 1.000 | 1.000 | 1.000 | 15 | 33.33 | 66.67 |
| CWE-388 | 1.000 | 1.000 | 1.000 | 1.000 | 36 | 8.33 | 91.67 |
| CWE-22 | 1.000 | 1.000 | 1.000 | 1.000 | 74 | 4.05 | 95.95 |
| CWE-704 | 1.000 | 1.000 | 1.000 | 1.000 | 80 | 1.25 | 98.75 |
| CWE-254 | 1.000 | 0.933 | 0.966 | 0.997 | 302 | 4.97 | 95.03 |
| CWE-415 | 1.000 | 1.000 | 1.000 | 1.000 | 102 | 13.73 | 86.27 |
| CWE-369 | 1.000 | 1.000 | 1.000 | 1.000 | 78 | 6.41 | 93.59 |
| CWE-79 | 1.000 | 1.000 | 1.000 | 1.000 | 84 | 3.57 | 96.43 |
| CWE-404 | 0.500 | 0.667 | 0.571 | 0.963 | 81 | 3.70 | 96.30 |
| CWE-134 | 0.500 | 1.000 | 0.667 | 0.983 | 60 | 1.67 | 98.33 |
| CWE-346 | 1.000 | 1.000 | 1.000 | 1.000 | 6 | 16.67 | 83.33 |
| CWE-17 | 1.000 | 1.000 | 1.000 | 1.000 | 72 | 4.17 | 95.83 |
| CWE-77 | 1.000 | 1.000 | 1.000 | 1.000 | 22 | 9.09 | 90.91 |
| CWE-269 | 1.000 | 0.667 | 0.800 | 0.988 | 86 | 3.49 | 96.51 |
| CWE-611 | 1.000 | 1.000 | 1.000 | 1.000 | 46 | 6.52 | 93.48 |
| CWE-19 | 1.000 | 0.714 | 0.833 | 0.973 | 73 | 9.59 | 90.41 |
| CWE-617 | 1.000 | 1.000 | 1.000 | 1.000 | 99 | 4.04 | 95.96 |
| CWE-494 | 1.000 | 1.000 | 1.000 | 1.000 | 7 | 14.29 | 85.71 |
| CWE-287 | 1.000 | 1.000 | 1.000 | 1.000 | 59 | 6.78 | 93.22 |
| CWE-834 | 1.000 | 1.000 | 1.000 | 1.000 | 40 | 5.00 | 95.00 |
| CWE-665 | 1.000 | 1.000 | 1.000 | 1.000 | 8 | 37.50 | 62.50 |
| CWE-674 | 1.000 | 1.000 | 1.000 | 1.000 | 4 | 25.00 | 75.00 |
| CWE-668 | 1.000 | 1.000 | 1.000 | 1.000 | 7 | 14.29 | 85.71 |
| CWE-918 | 1.000 | 1.000 | 1.000 | 1.000 | 5 | 20.00 | 80.00 |
| CWE-682 | 0.750 | 1.000 | 0.857 | 0.900 | 10 | 30.00 | 70.00 |
| CWE-191 | 1.000 | 1.000 | 1.000 | 1.000 | 10 | 10.00 | 90.00 |
| CWE-18 | 1.000 | 1.000 | 1.000 | 1.000 | 5 | 80.00 | 20.00 |
| CWE-16 | 1.000 | 1.000 | 1.000 | 1.000 | 7 | 14.29 | 85.71 |
| CWE-824 | 1.000 | 1.000 | 1.000 | 1.000 | 1 | 100.00 | 0.00 |
| Overall | 0.946 | 0.940 | 0.943 | 0.993 | 18864 | 5.88 | 94.12 |

Table 14: Complete BigVul Metrics for All CWEs with Class Balance

| CWE | Accuracy | Total | Pos. % | Neg. % |
|---|---|---|---|---|
| CWE-601 | 0.875 | 8 | 0.00 | 100.00 |
| CWE-347 | 1.000 | 3 | 0.00 | 100.00 |
| CWE-426 | 1.000 | 3 | 0.00 | 100.00 |
| CWE-361 | 1.000 | 4 | 0.00 | 100.00 |
| CWE-285 | 1.000 | 25 | 0.00 | 100.00 |
| CWE-290 | 1.000 | 8 | 0.00 | 100.00 |
| CWE-94 | 0.875 | 8 | 12.50 | 87.50 |
| CWE-281 | 1.000 | 5 | 0.00 | 100.00 |
| CWE-706 | 1.000 | 1 | 0.00 | 100.00 |
| CWE-862 | 1.000 | 2 | 0.00 | 100.00 |
| CWE-693 | 1.000 | 7 | 0.00 | 100.00 |
| CWE-295 | 1.000 | 7 | 0.00 | 100.00 |
| CWE-1021 | 1.000 | 5 | 0.00 | 100.00 |
| CWE-255 | 1.000 | 4 | 0.00 | 100.00 |
| CWE-129 | 1.000 | 7 | 0.00 | 100.00 |
| CWE-120 | 1.000 | 8 | 0.00 | 100.00 |
| CWE-352 | 1.000 | 4 | 0.00 | 100.00 |
| CWE-327 | 1.000 | 1 | 0.00 | 100.00 |
| CWE-909 | 1.000 | 7 | 0.00 | 100.00 |
| CWE-74 | 1.000 | 1 | 0.00 | 100.00 |
| CWE-330 | 1.000 | 3 | 0.00 | 100.00 |
| CWE-90 | 1.000 | 4 | 0.00 | 100.00 |
| CWE-770 | 1.000 | 6 | 0.00 | 100.00 |
| CWE-172 | 1.000 | 1 | 0.00 | 100.00 |
| CWE-354 | 1.000 | 1 | 0.00 | 100.00 |
| CWE-502 | 1.000 | 2 | 0.00 | 100.00 |
| CWE-755 | 1.000 | 2 | 0.00 | 100.00 |
| CWE-664 | 1.000 | 2 | 0.00 | 100.00 |

NOTE: Some metrics not applicable due to single-class predictions

Table 15: Single Class CWE Results

