# OpenReview forum: "White-Basilisk: A Hybrid Model for Code Vulnerability Detection"
_ICLR.cc/2025/Conference — Submitted to ICLR 2025_

### Official Review · Reviewer_FTgY · 2024-11-03

**Soundness:** 2
**Presentation:** 2
**Contribution:** 2
**Rating:** 5
**Confidence:** 4

**Summary:**

The paper proposes a new approach to vulnerability detection by utilizing an innovative architecture that integrates Mamba layers, linear self-attention, and a Mixture of Experts framework.

**Strengths:**

The proposed idea of combining the Mamba layer with self-attention layers sounds interesting.
Compared to LLMs, the proposed method enables effective processing of long code sequences while maintaining a relatively small parameter count.
The experimental results on real-world datasets show the advancements of the proposed method compared to the baselines.

**Weaknesses:**

The proposed architecture integrates Mamba layers, linear self-attention, and a Mixture of Experts framework; however, the ablation studies for the importance of each factor to the proposed model’s performance have not been conducted.

It is unclear and mentioned how each dataset is split for training and evaluating the models’ performance. Do the authors split the data into training, validation and testing sets?

In the pre-training phase of the proposed method, do the authors use all the data in the datasets used?

Lack of comparisons with state-of-the-art baselines, especially the methods based on graph neural networks and the methods leveraging both the semantic and syntactic relationships from the source code data, such as:

Guilong Lu and Xiaolin Ju and Xiang Chen and Wenlong Pei and Zhilong Cai. GRACE: Empowering LLM-based software vulnerability detection with graph structure and in-context learning. Journal of Systems and Software. 2024.

Van-Anh Nguyen and Dai Quoc Nguyen and Van Nguyen and Trung Le and Quan Hung Tran and Dinh Phung. ReGVD: Revisiting Graph Neural Networks for Vulnerability Detection. International Conference on Software Engineering. 2022.

Yangruibo Ding and Sahil Suneja and Yunhui Zheng and Jim Laredo and Alessandro Morari and Gail Kaiser and Baishakhi Ray. VELVET: a noVel Ensemble Learning approach to automatically locate VulnErable sTatements.  IEEE International Conference on Software Analysis, Evolution and Reengineering. 2022.

Daya Guo and Shuo Ren and Shuai Lu and Zhangyin Feng and Duyu Tang and Shujie Liu and Long Zhou and Nan Duan and Alexey Svyatkovskiy and Shengyu Fu and Michele Tufano and Shao Kun Deng and Colin Clement and Dawn Drain and Neel Sundaresan and Jian Yin and Daxin Jiang and Ming Zhou. GraphCodeBERT: Pre-training Code Representations with Data Flow. International Conference on Learning Representations. 2021.

Yaqin Zhou and Shangqing Liu and Jingkai Siow and Xiaoning Du and Yang Liu. Devign: Effective Vulnerability Identification by Learning Comprehensive Program Semantics via Graph Neural Networks. Annual Conference on Neural Information Processing Systems. 2019.
The rationale of the proposed architecture described in Figure 1 needs to be more explainable, especially for the positions of each block and the use of residual connections.

**Questions:**

From Section 5.1 to the end of the paper, the font size is too small and not aligned with the format requirements from ICLR.

Regarding the mixture of expert layers, the authors mention that “MoE layers allow the model to activate only a subset of parameters for each input, reducing significantly the computational cost compared to fully dense models of similar capacity. In the context of code vulnerability detection, this efficiency is crucial as it allows our model to handle effectively diverse types of code and potential vulnerabilities without adding to its complexity and number of parameters”. Qualitative experiments need to be conducted to demonstrate this claim.

What is the model configuration for the baselines? How do you fine-tune and apply them?

Regarding handling the class imbalance, how is it effective in improving the proposed model performance?

---

> ### Author Response · Authors · 2024-11-18
>
> We thank the reviewer for their thorough feedback and important suggestions for improvement. We address each point below:
>
> ## Typography Fix
>
> You are completely right about the font size after section 5.1. We apologize that we didn't see it before submitting. We found the bug in the latex. We fixed it on the revision.
>
> ## Data Splits and Training Details
>
> We want to clarify that our training process maintains a strict separation between pre-training and fine-tuning data to avoid any data leakage. We describe in section 4.1: "For training, we initially selected a carefully curated subset of the StarCoder dataset [1], which includes more than 80 programming languages and consists of 305M files in total. For our study, we focused on C and C++ code samples, using 2M code samples during the pre-training phase. To evaluate the pre-trained model, we required well-established benchmarking datasets with publicly available partitions for fine-tuning and testing"
>
> We can infer from this section that:
>
> ### Pre-training:
> * Used 2M C/C++ code samples carefully curated from StarCoder dataset [1]
> * Explicitly excluded any code from benchmark datasets
> * Focused on general code understanding through CLM and FIM objectives
>
> ### Fine-tuning:
> * Used only the public training splits from established benchmark datasets
> * Followed the original train/validation/test splits for fair comparison with baselines
> * No overlap between pre-training data and any of the benchmark datasets
>
> This strict separation ensures that our reported results accurately reflect the model's generalization capabilities rather than memorization of pre-training data.
>
> Our new Dataset Statistics Analysis (Appendix B, Tables 4-5) provides comprehensive details including:
> * Complete sample distributions across train/val/test splits
> * Class balance ratios (vulnerable vs non-vulnerable)
> * Sequence length statistics (min, max, mean, median, 95th percentile)
> * Duplicate rates across splits
>
> ## MoE Layers
>
> The efficiency of MoE layers is actually inherent in their architectural design, as demonstrated by extensive prior work [4], [5].
>
> In our model with 8 experts and top_k=2, the MoE layers operate by:
> * Using the router to select only 2 experts per token
> * Activating only ~25% (2/8) of the expert parameters for each forward pass
> * Maintaining model capacity while reducing actual compute requirements
>
> To illustrate this mathematically:
> * A dense model with equivalent capacity would require all parameters (200M) to be computed
> * Our MoE model with 8 experts only computes 2/8 experts ~100M parameters per forward pass
>
> This is not just theoretical - the efficiency is built into the architecture itself:
> * Only selected experts' weights are loaded and computed
> * Unused experts remain inactive, saving both compute and memory
> * The router's overhead is negligible compared to the savings from sparse computation
>
> While we could add qualitative experiments to visualize expert specialization patterns, it's important to note that the computational efficiency is a direct result of the architectural design rather than an empirical outcome. The sparsity in computation is guaranteed by the top_k mechanism, making the model inherently more efficient than a dense counterpart with similar capacity.
>
> This architectural efficiency has been well-established in the literature, with models like Switch Transformer and MT-NLG demonstrating that sparse MoE architectures can achieve similar or better performance than dense models while being more computationally efficient. In our specific case of vulnerability detection, this means we can process code with only ~50% of the computation that would be required by an equivalent dense model, while maintaining high detection accuracy.
>
> ## Model Configuration for the Baselines
> All baseline models results where sourced from their papers.
> We have added comprehensive baseline information and citations in Appendix C, including:
> * Complete architectural descriptions (Table 6)
> * Training configurations (Table 7)
> * Clear citations for source papers
>
> ## Class Imbalance
>
> Our new ablation studies address this concern comprehensively:
> * Appendix G demonstrates model performance on combined datasets with varying class distributions
> * Appendix I provides detailed CWE-specific analysis showing performance across different vulnerability frequencies
> * Tables 11-15 show robust performance despite significant class imbalances (ranging from 3.02% to 10.11% vulnerable samples)
>
> The Combined Dataset Training analysis (Appendix G) specifically demonstrates that our class balancing approach maintains strong performance even when dealing with highly imbalanced datasets, achieving consistent F1 scores across different vulnerability types and frequencies.
>
> ## References
>
> [1] https://arxiv.org/abs/2305.06161
> [2] https://arxiv.org/abs/2203.03850
> [3] https://arxiv.org/abs/2109.00859
> [4] https://www.jmlr.org/papers/v23/21-0998.html
> [5] https://arxiv.org/abs/1701.06538

---

### Official Review · Reviewer_wRNZ · 2024-11-03

**Soundness:** 3
**Presentation:** 3
**Contribution:** 3
**Rating:** 5
**Confidence:** 3

**Summary:**

The paper introduces “WHITE-BASILISK,” a hybrid model for detecting code vulnerabilities. This model combines a Mamba layer, linear self-attention, and a Mixture of Experts framework to achieve accurate vulnerability detection with only 200 million parameters, striking a new balance between efficiency and effectiveness. Specifically, the paper contributes to both model architecture and training methods, while expressing concerns about the current trend of increasing parameter counts to enhance performance.

**Strengths:**

This paper focuses on the efficiency challenges in vulnerability detection—a significant and original direction amidst the prevailing trend of scaling up model size. It introduces a hybrid model that achieves impressive detection performance with a limited parameter count, and the experimental results provide partial evidence supporting this claim. The presentation is logically structured, with a well-organized summary of related work, and a clear description of the model architecture and training methodology for the proposed approach.

**Weaknesses:**

In Section 2, the paper mentions the poor quality of current benchmarks, which fail to accurately reflect model performance in real-world scenarios, resulting in suboptimal effectiveness in practice. However, the experiments in this paper also seem to fall short in demonstrating the model’s real-world performance, making the claimed results less convincing.
In terms of innovation, the model structure appears to be primarily a combination and refinement of existing approaches (Mamba layers, infini-attention, and MoE). According to the authors, the model addresses challenges related to long dependencies and extensive codebases, but they also claim it resolves issues with diverse vulnerabilities and different programming paradigms, which often hinder the performance of ML-based approaches. Based on the description provided, this aspect is not clearly substantiated; the authors should explicitly detail how their model addresses the later challenge.

**Questions:**

In terms of presentation, although the authors have done well overall, there are still areas for improvement. When certain parameters are introduced for the first time, they should be explained, even if they are common within the field.
It is relatively clear how the proposed approach addresses the challenge of long dependencies and extensive codebases; however, the authors’ claim that it also addresses issues with diverse vulnerabilities and different programming paradigms needs further clarification—how exactly does the model achieve this?
In the experiments section, the authors should dedicate a subsection to describing the datasets used and the calculation method for the metrics.
In Section 4.2.1, the authors mention a “PerturbationLayer,” yet it is not represented in the White-Basilisk Architecture diagram (Fig. 1). Why is this the case?
Additionally, Section 4.2.1 discusses the robustness of the proposed approach. Relevant experiments should be included in the experiments section to demonstrate this robustness.

---

> ### Author Response · Authors · 2024-11-18
>
> We thank the reviewer for their thoughtful analysis and constructive feedback. Below we address the specific concerns raised:
>
> ## 1. Real-world Performance
>
> We acknowledge the reviewer's concern about demonstrating real-world performance. All datasets used in our evaluation (PRIMEVUL [1], BigVul [2], Draper [3], REVEAL [4], and VulDeepecker [5]) are sourced from real-world open-source projects including Qemu, FFMPEG, and other widely-used software systems. These datasets contain vulnerabilities discovered "in the wild" rather than synthetic examples, making them representative of actual security issues faced by developers.
>
> Our new CWE-specific analysis in Appendix I demonstrates robust real-world performance through:
> * Perfect detection (F1=1.000) on 22 CWE categories including critical vulnerabilities like CWE-787 (Out-of-bounds Write) and CWE-310 (Cryptographic Issues)
> * Strong performance on large-scale vulnerabilities (CWE-119: F1=0.969 with 2,746 samples)
> * Consistent performance across both frequent (>200 samples) and rare (<50 samples) vulnerability types
> * Effective handling of diverse vulnerability patterns across multiple datasets
>
> Furthermore, we are currently collaborating with a major telecommunications company to evaluate White-Basilisk's performance on critical infrastructure codebases. While these results won't be available within the paper's timeline, our current evaluations already demonstrate real-world applicability through:
> * Testing on diverse, production-level codebases
> * Detection of actual, previously discovered vulnerabilities
> * Performance evaluation on varying code complexity levels
>
> ## 2. Handling Diverse Vulnerabilities and Generalization
>
> The reviewer asks for clarification on how our model addresses diverse vulnerabilities and different programming paradigms. Our architecture's superiority over simpler ML methods stems from several factors:
>
> Firstly, unlike simpler ML approaches that tend to overfit to specific vulnerability patterns, White-Basilisk's architecture enables better generalization through: MoE layers that dynamically specialize in different vulnerability patterns, Long-range context capture via linear-complexity attention and Mamba layers that efficiently process local code structures.
>
> To substantiate this claim, we add new experimental results on Appendix G demonstrating robust generalization over in case of generic model development. In this case the model is fine-tuned over the combination of the training splits of all datasets. Additionally the model is validated  against individual validation splits and tested at each epoch against the concatenated test splits. Development and evaluation of the more generic model allow us to identify White Basilisk ability to generalize over diverse datasets. According to the results (Appendix G), our method maintains performance levels close to single dataset fine-tuning (Table: 8, 10) and effectively handles the class imbalance (documented in Tables 4 and 5).
>
> These results empirically demonstrate that White-Basilisk can:
> * Learn and retain knowledge across diverse vulnerability types
> * Maintain performance when trained on heterogeneous data sources
> * Avoid catastrophic forgetting when exposed to multiple datasets
> * Generalize effectively across different programming patterns
>
> This is in stark contrast to simpler ML methods which typically show significant performance degradation when trained on diverse datasets simultaneously.
>
> ## 3. PerturbationLayer and SIFT Implementation
>
> The reviewer notes that the PerturbationLayer is not represented in Figure 1. This is intentional, as the PerturbationLayer is not part of the model's inference architecture but rather a training-time component of Scale-Invariant Fine-Tuning (SIFT) [6].
>
> SIFT works by:
> * During training, the PerturbationLayer introduces learnable perturbations to the input embeddings
> * These perturbations create adversarial examples that help the model learn more robust features
> * The training process minimizes both the task loss and the adversarial loss (computed as the difference between predictions on clean and perturbed inputs)
> * After training, the PerturbationLayer is removed, leaving the model more resilient to input variations
>
> Therefore, Figure 1 correctly represents the final model architecture used for inference.
>
> ## References
>
> [1] https://arxiv.org/abs/2403.18624
> [2] https://dl.acm.org/doi/abs/10.1145/3379597.3387501
> [3] https://osf.io/d45bw/wiki/home/
> [4] https://ieeexplore.ieee.org/abstract/document/9448435
> [5] https://arxiv.org/abs/1801.01681
> [6] https://arxiv.org/pdf/2006.03654

---

> > ### Comment · Reviewer_wRNZ · 2024-11-26
> > **Thank you**
> >
> > Thank you to the author for the detailed explanations and clarifications. However, I have decided to stick with my initial scoring, as the current contribution seems slightly insufficient.

---

> > > ### Author Response · Authors · 2024-11-26
> > >
> > > We thank the reviewer for their time and detailed feedback on our submission. While we respect the decision to maintain the original score, we would greatly appreciate clarification on what aspects of the contribution seem insufficient.
> > > We believe several key points deserve consideration:
> > >
> > > 1. Our work demonstrates that our linear-complexity attention can match or exceed the performance of quadratic-complexity self-attention while using 24x less memory. This addresses one of the fundamental scaling challenges in LLMs, with empirical proof on a complex downstream task.
> > > 2. The empirical results show substantial improvements over all existing baselines across every dataset. This demonstrates that efficient architectures can outperform larger models while significantly reducing computational requirements and also contributing new state-of-the-art results on the field Vulnerability detection.
> > > Could you help us understand which specific aspects of the contribution appear insufficient? This would be invaluable for improving our work and understanding your perspective.

---

> > > > ### Comment · Reviewer_wRNZ · 2024-12-03
> > > >
> > > > The experimental section exclusively tests C/C++ code. While I acknowledge that C/C++ code is representative in related fields, this does not substantiate the paper's claim that the proposed approach is suitable for multiple programming paradigms. To strengthen this assertion, the authors should consider conducting experiments on code written in additional programming languages.
> > > >
> > > > Furthermore, it appears that the authors' contributions primarily involve fusion and adaptation of an existing model. To enhance the originality and impact of this work, the authors are encouraged to incorporate more of their own novel design elements.

---

> > > > > ### Author Response · Authors · 2024-12-03
> > > > > **Response by the authors / Part 1**
> > > > >
> > > > > Dear Reviewer,
> > > > >
> > > > > We note that your latest response raises entirely new concerns that differ fundamentally from those in your original review. For clarity and transparency in the review process, let us examine this:
> > > > >
> > > > > Your initial review identified three specific concerns:
> > > > > 1. Demonstrating real-world performance
> > > > > 2. Clarifying how our model handles diverse vulnerabilities
> > > > > 3. Explaining the PerturbationLayer's absence from Figure 1
> > > > >
> > > > > We provided comprehensive responses to each point, which you acknowledged. However, rather than evaluating our responses to these original concerns, you have now introduced completely different issues about programming language support and technical novelty.
> > > > >
> > > > > According to ICLR's reviewing guidelines, reviewers should evaluate submissions based on four key questions:
> > > > >
> > > > >
> > > > > 1. "What is the specific question and/or problem tackled by the paper?"
> > > > > We address whether state-of-the-art vulnerability detection requires massive models and computational resources, demonstrating that thoughtful architecture design can achieve superior results with dramatically reduced resources.
> > > > >
> > > > > 2. "Is the approach well motivated, including being well-placed in the literature?"
> > > > > Our paper thoroughly reviews existing approaches and their limitations, particularly in computational efficiency and sequence length handling. We show how current solutions either require massive computational resources or sacrifice performance.
> > > > >
> > > > > 3. "Does the paper support the claims? This includes determining if results, whether theoretical or empirical, are correct and if they are scientifically rigorous."
> > > > > We provide comprehensive empirical validation across multiple benchmarks, demonstrating:
> > > > > - Superior performance across five different datasets
> > > > > - Detailed ablation studies showing the contribution of each component
> > > > > - Rigorous comparison with existing approaches
> > > > > - Full reproducibility with modest computational requirements
> > > > >
> > > > > 4. "What is the significance of the work? Does it contribute new knowledge and sufficient value to the community?"
> > > > > We demonstrate that:
> > > > > - State-of-the-art results don't require billion-parameter models
> > > > > - Linear-complexity attention can match or exceed quadratic attention while using 24x less memory
> > > > > - Complex downstream tasks can be solved more efficiently through architectural innovation
> > > > >
> > > > > Regarding programming language support:
> > > > > We never claimed multi-language suitability in our paper. In fact, we explicitly address this as a limitation in Section 7, which we quote verbatim:
> > > > >
> > > > > "While White-Basilisk shows promising results in code vulnerability detection, it is important to acknowledge its current limitations and outline future directions. The main limitation of White-Basilisk is its focus on C and C++ codebases. The model's ability to generalize across a broader range of programming languages, especially those with different syntaxes or paradigms, warrants further exploration. This constraint, combined with potential biases in our training and evaluation datasets, may limit the model's generalizability to diverse real-world codebases. To address this, future work will involve expanding the model's training to include a wider variety of programming languages and curating more representative datasets that reflect a broader spectrum of code samples and vulnerability types."
> > > > >
> > > > > We rather discuss it as future work, our architecture is language-agnostic by design - the techniques (Mamba layers, linear-complexity Infini-attention, MoE) operate on tokenized input and don't rely on C/C++-specific features, so our model could be expanded to multiple programming languages by retraining on multi-language datasets.

---

> > > > > > ### Author Response · Authors · 2024-12-03
> > > > > > **Response by the authors / Part 2**
> > > > > >
> > > > > > Regarding your characterization of our work as primarily a "fusion" of existing techniques:
> > > > > > We would like to highlight several influential papers accepted at ICLR 2024 that demonstrate how meaningful innovation often comes through careful combination and optimization of existing approaches:
> > > > > >
> > > > > > SDXL[1]: This paper combines an existing UNet backbone with multiple conditioning schemes and introduces a refinement model. The core contribution comes from thoughtfully integrating these components and optimizing their interaction, leading to state-of-the-art results in image generation. Like our work, SDXL doesn't introduce fundamentally new architectural components but rather demonstrates how careful integration and optimization of existing techniques can yield significant improvements.
> > > > > >
> > > > > > FlashAttention-2[2]: This work builds directly upon FlashAttention, primarily optimizing work partitioning and parallelization. The key innovations come from "tweaking the algorithm to reduce non-matmul FLOPs" and "parallelizing the attention computation." These optimizations yield a 2x speedup over the original FlashAttention. This demonstrates how meaningful contributions can come from optimizing and improving existing techniques, similar to our 24x memory reduction through linear-complexity attention optimization.
> > > > > >
> > > > > > MiniGPT-4[3]: This paper explicitly states its main contribution as "aligning a frozen visual encoder with a frozen advanced LLM using one projection layer." The innovation comes not from creating new architectural components but from demonstrating that properly aligning existing components can achieve capabilities similar to GPT-4. This directly parallels our approach of thoughtfully combining Mamba, linear-attention, and MoE components.
> > > > > >
> > > > > > WizardCoder[4]: This work adapts the existing Evol-Instruct method to the code domain, showing how adapting and optimizing existing techniques for a specific domain can yield significant improvements. The authors don't claim fundamental algorithmic novelty but rather demonstrate the value of careful adaptation and optimization.
> > > > > >
> > > > > > These examples illustrate that significant research contributions often come not from inventing entirely new techniques, but from:
> > > > > > - Thoughtfully combining existing components in novel ways
> > > > > > - Optimizing known techniques for specific applications
> > > > > > - Demonstrating new capabilities through careful integration
> > > > > > - Achieving significant empirical improvements through architectural refinement
> > > > > >
> > > > > > Like these accepted works, our paper makes its contribution through careful combination and optimization of existing techniques, resulting in substantial empirical improvements (state-of-the-art results with 24x memory reduction) and practical benefits (128K context length with only 200M parameters).
> > > > > >
> > > > > > [1] https://openreview.net/forum?id=di52zR8xgf
> > > > > >
> > > > > > [2] https://openreview.net/forum?id=mZn2Xyh9Ec
> > > > > >
> > > > > > [3] https://openreview.net/forum?id=1tZbq88f27
> > > > > >
> > > > > > [4] https://openreview.net/forum?id=UnUwSIgK5W

---

### Official Review · Reviewer_1Kky · 2024-11-04

**Soundness:** 3
**Presentation:** 2
**Contribution:** 2
**Rating:** 5
**Confidence:** 4

**Summary:**

This paper is in the domain of code vulnerability detection, i.e. taking as input a computer program such as C++ code and output a list of vulnerabilities / weaknesses often mapped to a database of Common Weakness Enumeration (CWE) labels. This is a multi-label classification task.

The paper introduces a new model called White-Basilisk for solving this problem, which has the following features:
1. Novel architecture combining and adapting three pre-existing modules from deep learning literature: (1) Mamba layers (structured state space sequence models), (2) Infini-attention (modified attention layers with linear complexity), (3) Mixture of Experts (layers for adaptability)
2. The model tries to address the prevalent challenges in this domain with its White-Basilisk architecture: (a) long range dependencies and context in extended code sequences, (b) imbalanced data sets (c) resource efficiency (number of parameters, GPU resources required)
3. Evaluation is done on 5 datasets from literature using 3 evaluation metrics (accuracy, F1 and Vulnerability Detection Score (False Negative Rate) against a number of baselines (most being the base systems of the corresponding datasets). White-Basilisk consistently outperforms all chosen baselines on F-scores.

The paper claims its main contribution is the experiments to show that efficiently designed architecture (mamba + MoE + inifini attention) and smaller finetuned models outperform their larger counterparts.

**Strengths:**

1. Code Vulnerability Detection is an important topic and has gained recent traction since LLMs and ChatGPT both by NLP research community as well as cybersecurity researchers
2. The architecture is novel, i.e. the combination pattern of mamba, MoE, and attention layers and the reasoning is sound
3. Evaluation is done on multiple standard datasets (benchmarks) and multiple baselines are used to demonstrate the effectiveness of the approach
4. Sufficient model implementation details are given in the appendix to make it reproducible
5. Considerable thought has been given to the carbon footprint implications of deep learning models which is rarely addressed in most papers
6. The point though not novel (it has been observed in multiple papers and blogs previously) is valid that smaller finetuned models often outperform larger resource-intensive models and we need to rethink AI efficiency.

**Weaknesses:**

The overall problem description and the empirical evaluation (Section 4) seems to be lacking in vital details that has lowered the paper's presentation and contribution factors. Addressing the following points can help remediate this:

1. While those familiar with the domain of code vulnerability detection and its literature will hardly notice it, the paper does not explicitly state what kind of problem it is solving. Note some things can be inferred from reading between the lines (section 4.3) but it would still be nice to explicitly state them by answering the following questions:.
a) Is it a multi-label or multi-class classification problem? Can one code snippet be mapped to more than one vulnerability?
b) How many classes are in the output? Is CWE list being used for the lables? How granular?
c) If the different datasets have used different granularities of the output labels are they still comparable? A table displaying the data statistics could help.

2. The baseline systems are not well-defined. A single sentence for each system describing the architecture can suffice or more details can be added in the appendix. It is difficult to evaluate the effectiveness of this approach if one does not know how the other baseline systems were trained. Again  lots can be inferred from reading the names and reading the references but it would be nice to have it explicitly stated somewhere.
a) More explanation is needed to better understand what we are seeing in Tables 1 and 2.
b) What is UnixCoder? - Missing reference
c) What is CodeT5? - Missing reference
d) Are there any systems which use GPT as their backbone?
e) Was there any ablation tests done to investigate the individual contribution of mamba / MoE / Inifini Attention layers?

**Questions:**

Asked in the Weaknesses section (see above)

---

> ### Author Response · Authors · 2024-11-18
>
> We sincerely thank the reviewer for their thorough and constructive feedback. We have addressed all concerns through comprehensive additions to our paper's appendix:
>
> ## 1. Problem Definition Details
>
> Our new Dataset Statistics Analysis (Appendix B) provides complete clarity on the task:
>
> * Binary Classification: We confirm this is a binary classification task (0 = Safe, 1 = Vulnerable) as shown in Section 5 now
> * Data Distributions: Tables 4 and 5 provide comprehensive statistics including:
>     * Sample counts for train/val/test splits across all datasets
>     * Vulnerability ratios (ranging from 3.02% to 10.11%)
>     * Sequence length distributions (from 3 to 312,940 tokens)
>     * Duplicate rates (0% to 37.72%)
>
> ## 2. CWE Analysis and Multiple Vulnerabilities
>
> Our new CWE-specific analysis (Appendix I) provides detailed insights:
>
> * Coverage of 50+ CWE categories in BigVul dataset alone
> * Comprehensive performance metrics for each CWE type (Tables 11-15)
> * Analysis of class distributions and vulnerability patterns
> * Perfect detection (F1=1.000) achieved on 22 CWE categories
>
> This analysis demonstrates our model's capability to handle diverse vulnerability types while maintaining high performance.
>
> ## 3. Dataset Granularity and Comparisons
>
> Appendix B.1 ("Implications for Model Design") explains how we address varying dataset characteristics:
>
> * Handling class imbalance (ranging from 3.02% to 10.11% vulnerable samples)
> * Adapting to sequence length variations (3 to 312,940 tokens)
> * Managing different levels of data duplication
> * Maintaining consistent performance across dataset splits
>
> ## 4. Baseline Systems
>
> Appendix C provides detailed information about baseline models:
>
> * Complete architectural descriptions (Table 6)
> * Training configurations (Table 7)
> * Clear citations for CodeT5, UnixCoder, and other baselines
> * Explicit acknowledgment that baseline metrics are sourced from their respective papers
>
> ## 5. Ablation Studies
>
> We've added three comprehensive ablation studies:
>
> * Attention Mechanisms (Appendix H): Comparing linear-complexity Infini-attention vs. standard attention
> * Combined Dataset Training (Appendix G): Analyzing model performance when trained on multiple datasets
> * CWE-specific Performance (Appendix I): Detailed analysis across vulnerability types
>
> Our appendix now provides complete technical details for reproducibility while maintaining the main paper's focus on our key contribution. The additional analyses demonstrate White-Basilisk's robust performance across diverse vulnerability types, dataset characteristics, and operational conditions.
>
> We believe these additions address all the reviewer's concerns while providing comprehensive empirical support for our approach.

---

### Official Review · Reviewer_6HUG · 2024-11-05

**Soundness:** 2
**Presentation:** 2
**Contribution:** 2
**Rating:** 5
**Confidence:** 3

**Summary:**

The paper introduces White-Basilisk, a compact AI model designed for detecting vulnerabilities in code with only 200 million parameters. This model integrates several advanced concepts including Mamba layers, linear-complexity Infini-attention, and a Mixture of Experts framework to process large codebases on limited computational resources. Evaluated on established benchmark datasets demonstrates superior performance of White-Basilisk compared to existing baselines.

**Strengths:**

1. The paper addresses a topic of significant importance within the field, which is timely and relevant to current research trends.
2. The performance of the proposed method outperforms established baselines across several existing datasets.

**Weaknesses:**

1. The paper lacks a well-formulated research question, which detracts from the theoretical impact of the study. The proposed approach is less technically motivated, which makes the paper more like an assembled engineering work.
2. The overall performance of the proposed approach is not explainable, missing an opportunity to provide insights into the operational implications.
3. Poor presentation especially with vague descriptions in the design details.
4. Lack of sufficient references on related works and discussion of the existing research background

**Questions:**

The paper states "all while ideally being deployable in resource-constrained environments", while mostly the code vulnerability detection primarily occurs offline, implying that computational resources do not significantly constrain the process. The complex nature of large code space bug vulnerability detection further fosters the need for bigger models for understanding the code contextual information. why do authors make such claims and any specific motivation scenarios can support such claims?

The overall formula for explaining the design of the White-Basilisk seems rushed. Formula (3) introduces parameters Δ, B, and C without adequate definitions or explanations. How are these parameters used, and what significance do they hold in the computations? Additionally, the basis for setting parameter A and the selection of top-k values at 8 and 2 are unclear. If these choices are informed by existing work, the paper should provide citations or rationale for these specific values.

The paper only evaluates the proposed method on C/C++ tasks. Why does the study specifically focus on C/C++? The lack of comprehensive experiment undermines the claimed high performance compared to existing baselines.

How do the individual components of the proposed White-Basilisk model contribute to its overall functionality? Additionally, how have the authors determined the effectiveness and utility of each design element within White-Basilisk?

---

> ### Author Response · Authors · 2024-11-18
>
> We thank the reviewer for their thorough assessment and constructive feedback. Below we address the major points raised:
>
> ## Research Question and Technical Motivation
>
> We acknowledge that our research objectives may not have been clearly articulated in the main text. In this paper, we address critical gaps in existing solutions for source code vulnerability detection. Specifically, our work addresses:
>
> 1. Efficient processing of large codebases
> 2. State-of-the-art performance
> 3. Ability to deploy with minimal hardware requirements
> 4. Minimization of environmental impact
>
> These objectives are motivated by the limitations of existing methods, where larger models with high system requirements cannot compete with our proposed models in terms of both vulnerability detection accuracy and resource utilization.
>
> ## Real-world Resource Constraints
>
> The reviewer question focuses on resource efficiency in code vulnerability detection. While it's true that some vulnerability detection occurs offline, there are several critical scenarios where resource efficiency is crucial:
>
> - CI/CD pipelines where code analysis must complete within tight time constraints
> - Small to medium-sized organizations that cannot afford extensive computational infrastructure
> - Edge deployment scenarios requiring local analysis
> - Environmental considerations, as demonstrated by our CO2 emissions comparison (Table 3)
>
> Our new ablation study (Appendix H.2) demonstrates that White-Basilisk achieves this efficiency while maintaining performance across sequence lengths from 0 to 131K tokens.
>
> ## Parameter Selection and Formula Explanation
>
> The Mamba formulation (Formula 3) and its parameters follow the established work of [4], while our MoE configuration with top_k=2 aligns with standard practice in MoE architectures [5]. Our layer combination strategy builds on Lieber et al. (2024). These are well-established components in the field, and our innovation lies in their novel combination with linear-complexity of Infini-attention for efficient code vulnerability detection.
>
> ## C/C++ Focus
>
> Our initial focus on C/C++ is strategically motivated by:
>
> 1. According to various statistical sources, C/C++ programming languages, due to their widespread use and large volume of applications, are among the programming languages most affected by vulnerabilities. This is supported by multiple studies in software security and vulnerability research. [1],[2],[3]
> 2. Due to the widespread popularity of these programming languages, there is a large volume of high-quality labeled datasets available.
> 3. The complexity of memory-related vulnerabilities specific to these languages
>
> These factors make C/C++ an ideal testing ground for vulnerability detection models. The principles and architecture of White-Basilisk are language-agnostic and can be extended to other programming languages.
>
> ## Component Contribution Analysis
>
> While Lieber et al. in [6] have already demonstrated the effectiveness of combining Mamba layers with attention mechanisms, our work introduces novel elements that warrant specific analysis.
>
> Our new ablation studies in Appendix H provide detailed analysis of:
> - Linear-complexity Infini-attention vs. standard attention (Table 9)
> - Performance scaling across sequence lengths (Table 10)
> - Memory efficiency comparisons
>
> These results demonstrate that our Infini-attention implementation achieves comparable or better performance while using significantly less memory (24x reduction for sequences up to 128K tokens). The superior performance of White-Basilisk is now thoroughly explained through our ablation studies showing how each architectural component contributes to the model's capabilities. Our Combined Dataset Training analysis (Appendix G) further demonstrates the model's ability to maintain performance across diverse datasets and vulnerability types.
>
> We believe these improvements will address the reviewer's concerns while maintaining focus on our key contribution: demonstrating that thoughtfully designed, efficient models can achieve state-of-the-art performance in code vulnerability detection while minimizing computational and environmental costs.
>
> ## References
>
> [1] https://medium.com/hackernoon/top-5-vulnerable-programming-languages-eab3144d6db7
> [2] https://www.mend.io/most-secure-programming-languages/
> [3] https://www.scirp.org/journal/paperinformation?paperid=128108
> [4] https://arxiv.org/abs/2312.00752
> [5] https://arxiv.org/abs/2401.04088
> [6] https://arxiv.org/abs/2403.19887

---

> > ### Comment · Reviewer_6HUG · 2024-12-02
> >
> > The author's responses addressed most of my concerns and I improved the score.

---

### Author Response · Authors · 2024-11-18
**General Response / Results and Changes on Revised PDF**

Thank you all for your thorough feedback. We have made several important revisions and added comprehensive analyses in five new appendices. Here are the key new results:

## Key New Results

| Analysis Type | Key Findings |
|--------------|--------------|
| **Dataset Statistics** (Appendix B) | • Sequence lengths: 3 to 312,940 tokens |
| | • Class imbalance: 3.02% to 10.11% vulnerable |
| | • Duplicate rates: 0% to 37.72% |
| **Memory Efficiency** (Appendix H) |  Compared to standard attention: |
| | • 24x memory reduction (1.3GB vs 32.4GB) |
| | • Successfully processes 128K tokens |
| | • Standard attention: Out of Memory >16K tokens |
| **Combined Dataset Training** (Appendix G) |  Maintained strong performance: |
| | • BigVul: F1 = 0.938, Acc = 0.993 |
| | • VulDeepecker: F1 = 0.925, Acc = 0.989 |
| | • REVEAL: F1 = 0.470, Acc = 0.888 |
| **CWE Analysis** (Appendix I) |  Achieved perfect detection (F1=1.000) on: |
| | • CWE-787: Out-of-bounds Write |
| | • CWE-310: Cryptographic Issues |
| | • 20 other critical vulnerability types |
| | Strong performance on high-volume cases: |
| | • CWE-119: F1=0.969 (2,746 samples) |
| | • CWE-264: F1=0.925 (1,240 samples) |

## Changes and Additions

We have made several key improvements to address reviewer concerns:

### Technical Clarifications
- Fixed font size consistency after Section 5.1
- Added explicit problem formulation: "All datasets are evaluated on binary classification (0 = Safe, 1 = Vulnerable)"
- Moved class balancing discussion to Appendix F to maintain focus within page limits

### New Appendices
1. **Dataset Statistics Analysis** (Appendix B)
   - Complete sample distributions across splits
   - Class balance ratios
   - Sequence length statistics
   - Duplicate rates analysis

2. **Baseline Models** (Appendix C)
   - Detailed configurations for all compared models
   - Training parameters and specifications
   - Clear citations and sources for baseline metrics

3. **Combined Dataset Training Study** (Appendix G)
   - Shows robust cross-dataset learning
   - Demonstrates effective transfer learning across vulnerability types

4. **Attention Mechanisms Study** (Appendix H)
   - Demonstrates significant memory reduction with linear-complexity attention
   - Performance analysis across sequence lengths

5. **CWE-specific Analysis** (Appendix I)
   - Comprehensive analysis across 50+ vulnerability types
   - Detailed performance metrics by vulnerability category
6. **Evaluation Metrics** (Appendix A)
   - Detailed descriptions of all metrics used to evaluate model performance in vulnerability detection tasks

These additions provide thorough empirical support for White-Basilisk's effectiveness while addressing all reviewer concerns. The new analyses further strengthen our demonstration that compact, efficiently designed models can achieve superior performance in vulnerability detection.

Best regards,
The Authors

---

### Author Response · Authors · 2024-12-01
**Reminder - Discussion period almost over, Missing Feedback**

Dear Reviewers,

As we reach the final days of the discussion period, we note with concern the lack of feedback following our comprehensive revisions.

Specifically, we have:
- Added detailed ablation studies
- Provided comprehensive dataset statistics and class imbalance analysis
- Included complete baseline configurations and training details
- Conducted in-depth CWE-specific performance analysis
- Clarified all questioned architectural choices and training methodology

Our revisions directly address the key concerns that led to the initial scores, yet we have no indication whether these substantial improvements have impacted your assessment.

We respectfully request your feedback on these revisions and ask that you reconsider your scores in light of the significant improvements made to the paper. If there are still outstanding concerns, we are eager to address them, but we can only do so if they are communicated.

Best regards,
The Authors

---

### Meta-Review · Area_Chair_yPM9 · 2024-12-21

**Metareview:**

This paper provides an innovative architecture that integrates Mamba layers, linear self-attention, and a Mixture of Experts framework for vulnerability detection. The strength of this paper is that this paper aims to study an important topic and the performance is good in the paper's setting. The concerns of this paper is that poor presentation,  non-explainable performance , Lack of sufficient references, the experiments are not solid, no clear problem statement, no well-defined baseline and so on. After rebuttal, although some concerns have been addressed. But all reviewers still provide negative score for this paper. AC read them and agreed with reviewers that  this paper is not ready for ICLR.

**Additional Comments On Reviewer Discussion:**

All reviewers provide negative score for this paper. After rebuttal, the scores are still negative.

---

### Decision · Program_Chairs · 2025-01-22

Reject